# Telomeres in ICF syndrome cells are vulnerable to DNA damage due to elevated DNA:RNA hybrids

Shira Sagie[1], Shir Toubiana[1,*], Stella R. Hartono[2,*], Hagar Katzir[1], Aya Tzur-Gilat[1], Shany Havazelet[1], Claire Francastel[3], Guillaume Velasco[3], Frédéric Chédin[2] & Sara Selig[1]

DNA:RNA hybrids, nucleic acid structures with diverse physiological functions, can disrupt genome integrity when dysregulated. Human telomeres were shown to form hybrids with the lncRNA TERRA, yet the formation and distribution of these hybrids among telomeres, their regulation and their cellular effects remain elusive. Here we predict and confirm in several human cell types that DNA:RNA hybrids form at many subtelomeric and telomeric regions. We demonstrate that ICF syndrome cells, which exhibit short telomeres and elevated TERRA levels, are enriched for hybrids at telomeric regions throughout the cell cycle. Telomeric hybrids are associated with high levels of DNA damage at chromosome ends in ICF cells, which are significantly reduced with overexpression of RNase H1. Our findings suggest that abnormally high TERRA levels in ICF syndrome lead to accumulation of telomeric hybrids that, in turn, can result in telomeric dysfunction.

[1] Molecular Medicine Laboratory, Rambam Health Care Campus and Rappaport Faculty of Medicine, Technion, Haifa 31096, Israel. [2] Department of Molecular and Cellular Biology and Genome Center, University of California, Davis, California 95616, USA. [3] Université Paris Diderot, Sorbonne Paris Cité, Epigenetics and Cell Fate, CNRS UMR7216, Paris Cedex 75205, France. * These authors contributed equally to this work. Correspondence and requests for materials should be addressed to S. Selig (email: seligs@tx.technion.ac.il).

Telomeres are nucleoprotein complexes that maintain the integrity and stability of eukaryotic chromosome ends[1,2]. T-loop formation, binding of shelterin and non-shelterin proteins and a unique chromatin structure, all protect the chromosome ends from deleterious events such as degradation and end-to-end fusions[3]. Human telomeres undergo gradual attrition with each cell division, and when they reach a critical length, telomere shortening elicits replicative senescence[4], with only a few short telomeres being sufficient to initiate this response[5,6]. In vertebrates, the telomeric DNA consists of a $(TTAGGG)_n$ repeat. Subtelomeres, the regions immediately proximal to telomeres, contain CpG-rich repetitive sequences, telomere-like repeats and families of larger repeats[7,8] and are packaged together with the telomeres as heterochromatin that assists in the stabilization of the chromosome ends[9]. Although sharing many characteristics, different human subtelomeres vary in size, sequence content and organization[10]. The transcription of telomeric repeat-containing RNA (TERRA), a long non-coding RNA produced by RNA polymerase II (refs 11,12), initiates within the subtelomeres at close proximity to the telomere tract and utilizes the telomeric C-rich leading strand as its template[7] (reviewed in ref. 13). TERRA has been implicated in numerous telomeric roles, such as regulation of telomere length, replication and heterochromatinization[14–19] (reviewed in refs 2,13). Evidence is emerging that the function and regulation of TERRA are telomere state dependent such that telomere length, telomerase expression and ALT pathway activity can influence the role that TERRA has at telomeres (reviewed in ref. 20).

R-loops, three-stranded nucleic acid structures that consist of a DNA:RNA hybrid and a displaced single-stranded DNA loop[21], are predisposed by strand asymmetry in the distribution of guanines and cytosines, termed GC-skewing. These structures form mainly co-transcriptionally when positive GC skew is present such that DNA:RNA hybrids form between the G-rich RNA strand and the C-rich complementary DNA strand[22]. Although various studies indicate that DNA:RNA hybrids have a positive effect on gene transcription and are beneficial to the cell[22–25], these structures have also been shown to mediate genome instability and replication stress[26]. R-loops have been implicated in human diseases, including trinucleotide expansion diseases, neurological diseases and cancer (reviewed in ref. 27). Telomeric DNA and TERRA transcripts are predicted to form hybrids, with the G-rich $(UUAGGG)_n$ TERRA transcript annealing to the C-rich $(CCCTAA)_n$ DNA template. Indeed, recent studies support the existence of such hybrids at telomeres in *S. cerevisiae* (whose telomeres are comprised of a different G-rich repeat)[14,28,29] and suggest that, in the absence of a telomere-maintenance mechanism, TERRA-telomeric DNA hybrids may promote accelerated telomere loss in *cis*[14]. In addition, telomeric DNA:RNA hybrids were found in various human cancer cells, both telomerase-positive and ALT cancerous cells[30]. In the latter, DNA:RNA hybrids were suggested to have a role in facilitating telomere recombination.

ICF (Immunodeficiency, Centromeric instability and Facial anomalies) syndrome type I is a rare autosomal-recessive syndrome, caused by hypomorphic mutations in the *DNMT3B* gene[31,32], the major DNA methyltransferase involved in *de novo* methylation of repetitive sequences in mammalian cells during development[32]. Subtelomeres, as other repetitive sequences, are severely hypomethylated in ICF type I syndrome cells[33–35]. We detected accelerated telomere shortening and significant telomere loss, premature replicative senescence and significantly elevated levels of TERRA transcripts in both ICF fibroblast and lymphoblastoid cells (LCLs)[33,35]. Although it was proposed that TERRA has a causative role in the generation of telomeric abnormalities in ICF syndrome[14,17,33–37], the

underlying mechanism by which this occurs is as yet unclear.

Here we further investigate the occurrence of human telomeric hybrids in various cell types. Furthermore, we address the question of whether all telomeres are equally competent in generating these hybrids and whether the subtelomeric regions may affect this capacity. Our findings establish that telomeric DNA:RNA hybrids occur also in primary human cells and that subtelomeric sequences have an effect on generation of telomeric hybrids. We demonstrate that elevated TERRA levels are associated with higher levels of telomeric hybrids in ICF syndrome and suggest a role for these DNA:RNA hybrids in promoting damage and instability at telomeric regions in this disease.

## Results

**Human subtelomeres are predicted to form DNA:RNA hybrids**. Human telomere-hexameric $(TTAGGG)_n$ repeats are predicted to form DNA:RNA hybrids, with the C-rich template annealing to the G-rich TERRA transcript. We validated this capacity *in vitro* and demonstrated, as in a previous study[30], that these hybrids are formed only in a specific direction and are sensitive to RNase H, an enzyme that specifically degrades RNA strands within DNA:RNA hybrids (Supplementary Fig. 1).

The majority of TERRA transcripts initiate at the last few hundred base-pairs (bps) of the subtelomeric region[7], although some TERRA species may start 5–10 kb upstream of the telomere tract[38]. As most DNA:RNA hybrids are assumed to form co-transcriptionally[22,39], we speculated that subtelomeric sequences might facilitate the formation of telomeric hybrids. To test this hypothesis, we first analysed the sequence of the distal 2 kb region adjacent to the telomere tract at both chromosome ends for CpG density, GC content and GC skew[23]. Regions with a strong GC skew downstream of the TERRA promoter may be prone to DNA:RNA hybrid formation. For this analysis, we utilized the previously described subtelomeric sequences[8,10], focussing on high-confidence subtelomeric regions whose sequence is available in the UCSC GRCh38/hg38 release with a clearly defined telomeric region or at least three consecutive TTAGGG repeats at the 3′ end. These subtelomeric regions were overlaid with the predicted TERRA promoters and transcription start sites (TSSs), as determined by the Genomatix software[40].

Most human subtelomeric regions exhibit high CpG density and GC content in regions corresponding to the predicted promoters for TERRA (Fig. 1a), thus closely resembling CpG island promoters. This is consistent with a similar analysis of a subgroup of TERRA promoters[7] and reinforced by the findings that TERRA transcribing telomeres show higher GC content in comparison to the non-transcribing ones[38]. Examination of GC skew revealed that its levels are variable over the entire 2 kb region. However, a substantial number, but not all, of the analysed subtelomeres possess positive GC skew in the region located immediately downstream of the predicted TERRA TSSs (Fig. 1a). Thus a subgroup of subtelomeres appears prone to form DNA:RNA hybrids at the end of chromosome arms, which could then extend into the neighbouring telomeric regions.

**Subtelomeric DNA:RNA hybrids form in human cell lines**. The analysis above prompted us to test whether DNA:RNA hybrids indeed form at distal human subtelomeric regions *in vivo*. Quantification of DNA:RNA hybrids is classically carried out by DNA:RNA hybrid immunoprecipitation (DRIP), a procedure that involves immunoprecipitation of these hybrids with the S9.6 antibody[22], and subsequently quantitative PCR (qPCR) or sequencing. For this, we analysed available genome-wide DRIP-

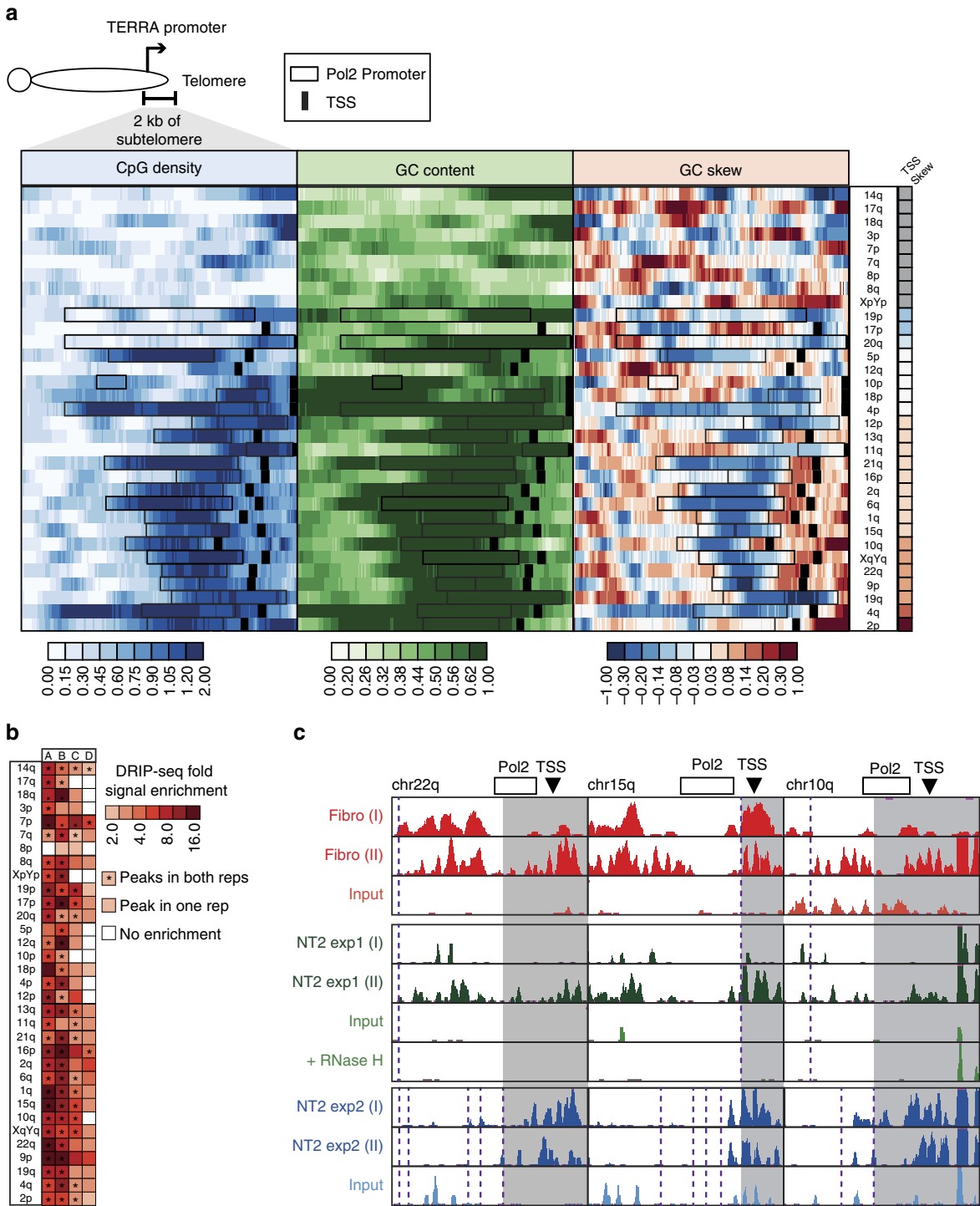

**Figure 1 | Sequence characteristics and potential for DNA:RNA hybrid formation in human distal subtelomeres.** (**a**) 2 kb distal subtelomeric regions immediately adjacent to the telomere repeat tract were analysed for CpG density, GC content and GC skew over tiled overlapping 200 bp windows. The value for each window is depicted using a colour heatmap, as indicated below; each tick mark corresponds to a window. The analysed subtelomeres, indicated on the right, are clustered by the GC skew downstream of the annotated TERRA promoter, when available (grey: no annotation). Predicted Pol2 promoters and putative TSSs are indicated by boxes and black vertical lines, respectively. (**b**) DRIP-seq signal over each subtelomeric region is indicated by a colour heatmap reflecting signal enrichment over input for each 2 kb window (colour scheme is indicated at the right). A, B, and C refer to three distinct DRIP-seq datasets in human fibroblast (Fibro - A) and human Ntera2 cells (NT2 - B and C) measured relative to input. Each dataset included two independent technical replicates. The presence of consistent signal peaks in each replicate within a dataset is noted by an asterisk (*), as indicated. D corresponds to C but measured relative to an RNase H-treated control. Subtelomeric regions are arranged in the same order as **a**. See 'Results' for details. (**c**) Representative screenshots of DRIP-seq data over three distinct subtelomeric regions (22q, 15q and 10q, as indicated). Normalized DRIP-seq signal densities are displayed for each region over two distinct replicates (indicated by (I) and (II)) as well as input and, when available, RNase H-treated controls. The position of the Pol2 promoters and putative TSSs is shown at the top. Vertical dashed lines indicate the position of restriction enzyme sites used to fragment the genomic DNA before DRIP-seq. The grey shaded area highlights a TRF encompassing the TERRA promoter and/or downstream regions showing significant DRIP-seq signal.

seq data sets[22,41,42]. One data set (referred to as Fibro in Fig. 1c) was generated from human primary fibroblasts[42]. Two independent data sets (referred to as NT2 exp1 and NT2 exp2 in Fig. 1c) were generated from the human embryonal carcinoma cell line Ntera2 (refs 22,41). Each dataset had two highly correlated technical replicates (Supplementary Fig. 2). Having established unequivocally that GC skew is a key determinant of DNA:RNA hybrid formation in both cell types (Supplementary Fig. 3), we proceeded to analyse these data sets for subtelomeric hybrid formation. As shown in Fig. 1b, many subtelomeres showed significant DRIP-seq signal enrichment over input in most data sets. In one instance, DRIP-seq signal enrichment in Ntera2 cells could be measured against an RNase H-treated control (referred to as NT2 exp1 and NT2 exp1 + RNase H in Fig. 1b). Again, most subtelomeric regions, in particular those predicted to be DNA:RNA hybrid-prone (Fig. 1a), showed significant enrichment for DRIP-seq signal, strongly suggesting the formation of DNA:RNA hybrids. Because the resolution of DRIP-seq is limited by the occurrence of restriction enzymes used to fragment the genome, we further explored the detailed patterns of DNA:RNA hybrid formation for evidence that the signal originates downstream of the TERRA promoter and TSS. In a number of instances, the restriction enzymes had cleavage sites in the immediate vicinity of the TERRA promoter, separating the TERRA promoter and downstream regions from upstream sequences (Fig. 1c). Strong DRIP-seq signals originating at or near the TERRA promoter and encompassing downstream subtelomeric regions adjacent to the telomere tract were observed for a number of subtelomeres (Fig. 1c). Thus the DRIP-seq data indicate that DNA:RNA hybrids can be generated at many chromosome ends. This phenomenon is not restricted to human cancer cells and occurs also in normal primary human cells.

**Telomeric DNA:RNA hybrids are elevated in ICF cells**. Primary human fibroblasts were shown previously to express low levels of TERRA[33]; however, these levels appear to be sufficient for hybrid formation at subtelomeric regions, as demonstrated above. As hybrid formation is correlated with transcript levels[26,43], we next asked whether the levels of DNA:RNA subtelomeric/telomeric hybrids would be influenced by TERRA levels. Cells from ICF type I syndrome patients provide an excellent platform to answer this question owing to their abnormally elevated TERRA levels. ICF cells were shown previously to display elevated levels of TERRA[33,34,37] and here we validate this finding for individual chromosome ends (Supplementary Fig. 4). We compared TERRA levels of five LCLs generated from ICF syndrome patients with four LCLs from normal individuals and three heterozygous carriers of ICF type I syndrome. We did not detect statistically significant differences between the normal and carrier LCLs throughout the various analyses; therefore, these samples were pooled together for further analyses and designated as the wild-type (WT) control group. Quantitative reverse transcription–PCR (qRT-PCR) analysis demonstrated that, while TERRA levels varied among the different subtelomeres, they were consistently higher in ICF compared with WT LCLs.

We then proceeded to determine the levels of DNA:RNA hybrid formation in ICF versus WT cells by DRIP analysis. Digested DNA from ICF and WT LCLs (Fig. 2a) was subjected to DRIP using the S9.6 hybrid-specific antibody, and the amount of immunoprecipitated material was compared with input using qPCR for 11 subtelomeric regions and 3 control non-telomeric regions (Fig. 2b,c and Supplementary Fig. 5a). No statistically significant differences were observed in the enrichment of hybrids at control regions between the WT and ICF groups, in agreement

with similar transcript levels of these genes (Supplementary Fig. 5b). Treatment of the DNA samples prior to DRIP with bacterial RNase H tested the specificity of the DRIP procedure, and, as expected, RNase H treatment significantly reduced or eliminated DRIP-qPCR signals emanating from both control and subtelomeric regions (Supplementary Fig. 5c,d). This analysis reveals that subtelomeric DNA:RNA hybrids form also in LCLs and that hybrid levels vary among subtelomeres both in the WT and the ICF groups (Fig. 2b,c). One-way analysis of variance for repeated measurements revealed a significant difference ($P$ value $< 0.05$) in the DRIP enrichment between the subtelomeres, for both the WT and ICF groups. Importantly, the amount of DNA:RNA hybrids at many of the tested subtelomeres was higher for ICF cells than for WT cells (Fig. 2b,c), suggesting that high TERRA expression leads to elevated formation of DNA:RNA hybrids at chromosome ends. DRIP enrichment levels of each individual LCL segregated the ICF from the WT group and six of the tested subtelomeric regions showed a statistically significant difference in hybrid formation between both groups (Fig. 2d). Five of the subtelomeric regions tested here did not show significant differences in DNA:RNA hybrid loads between WT and LCLs, even though lower average hybrid levels in WT cells were apparent for 10 of the 11 examined subtelomeres. Notably, the three subtelomeres that most substantially differed in TERRA levels between ICF and WT LCLs (2p, 10q and 15q) were also those that displayed significantly higher hybrid formation in ICF versus WT (Fig. 2d and Supplementary Fig. 4). Interestingly, the cumulative GC skew calculated downstream of the TERRA TSS/promoter segregated subtelomeres whose hybrid formation was affected by high TERRA levels in ICF cells from those that were not affected (Fig. 2e). Collectively, these data suggest that the formation of DNA:RNA hybrids at chromosome ends is influenced by the sequence characteristics of the region adjacent to the telomeric tract, including the TERRA promoter and TSS, and is promoted by elevated levels of TERRA and/or by perturbation of the normal chromatin structure in these regions in ICF syndrome cells.

**(TTAGGG)$_n$ are crucial for forming chromosome end hybrids**. As demonstrated above, the ability of human telomeric repeats to form DNA:RNA hybrids was confirmed *in vitro* by us (Supplementary Fig. 1) and another study[30]. Direct study of telomere repeat hybrids *in vivo* is impeded by the repetitive nature of the regions, which prevents their amplification by PCR. Therefore, DRIP analysis of terminal restriction fragments (TRFs), utilizing subtelomeric primers, does not distinguish whether hybrids are formed at telomere-repeats or subtelomeric regions or both (Fig. 2a). In order to allow this distinction, we searched for a common restriction site that would separate the telomeric tract from the subtelomeric region downstream to the TERRA promoter in several chromosome ends. We detected five chromosome ends (7q, 8p, 9p, 13q and 21q) in which *HinfI* restriction sites are properly positioned (Fig. 3a) and performed DRIP on these subtelomeres. In this set of experiments, we first digested all samples with the standard enzyme cocktail and then set aside one half of each sample for further digestion with *HinfI*. We then validated that the samples were digested efficiently, as shown in Supplementary Fig. 6 (boxed section of the figure). We analysed the paired samples ($+/-$ H) by DRIP-qPCR for four ICF LCLs (Fig. 3b,c) and four WT LCLs (Fig. 3d,e), including subtelomeres that encompass a *HinfI* site ($+$ *HinfI*) (Fig. 3b,d) and subtelomeres lacking a distal *HinfI* site between the hexameric repeat and the PCR-amplified region ($-$ *HinfI*) (Fig. 3c,e and Supplementary Fig. 7). The DRIP enrichment values obtained in the absence of *HinfI* digestion ($-$ H) were

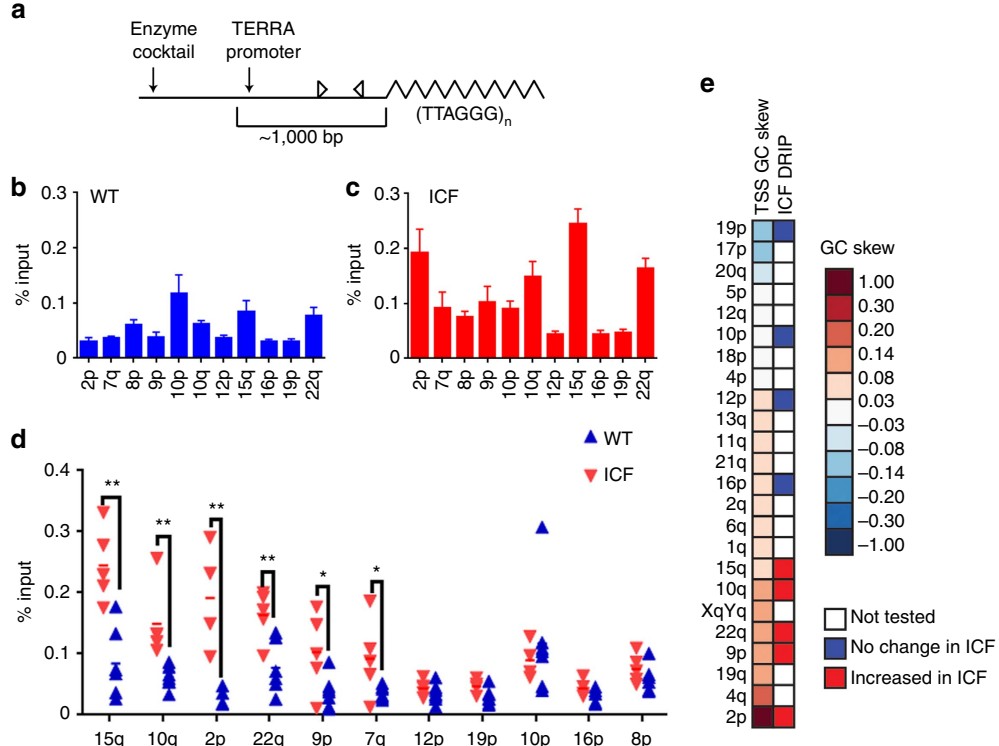

**Figure 2 | Telomeric DNA:RNA hybrids form preferentially in ICF cells.** (**a**) Schematic representation of the subtelomeric regions subjected to DRIP-qPCR analysis. Subtelomeric sequence appears as a straight line and telomeric TTAGGG repeats as a zigzagged line. The approximate 1,000 bp upstream to the telomere tract contains the TERRA promoter and TSS, and the two horizontal head-to-head arrows depict the location of the PCR primers used to amplify the DRIP sample. Predicted TERRA promoters are present in the majority of analysed subtelomeres within the TRF. (**b,c**) DNA extracted from five ICF and six WT LCLs was subjected to DRIP, followed by qPCR analysis of 11 subtelomeric amplicons representing 16 subtelomeric regions (Supplementary Table 4). The amount of immunoprecipitated material is shown as a percentage of input for each subtelomere for WT (**b**) and ICF samples (**c**). Bars and error bars represent means and s.e.m. calculated for all samples in each group over two to five independent experiments. (**d**) DRIP-qPCR values for each subtelomere are displayed for each LCL (WT samples—blue triangles, ICF samples—red triangles). Two-tailed Student's *t*-tests were performed to determine statistical differences between WT and ICF samples (** = *P* value < 0.01, * = *P* value < 0.05). Six out of the 11 subtelomeres studied showed significant differences. (**e**) GC skew downstream of TERRA promoter is predictive of increased DNA:RNA hybrid formation in ICF compared with WT LCLs. GC skew information is as shown in Fig. 1a. The telomeres that appear in this figure are those that contain a putative TSS.

arbitrarily set at 1, and the enrichment values obtained following *HinfI* digestion (+H) represent relative enrichment in comparison to the −H samples (Fig. 3b–e) (absolute percentage of input data are shown in Supplementary Fig. 6 and Supplementary Table 1).

In the ICF samples, at all of the subtelomeres containing a *HinfI* site close to the telomere tract, with the exception of 7q, a significant reduction in hybrid enrichment (*P* value < 0.001) was observed following removal of the telomeric sequence by *HinfI* digestion (Fig. 3b). This reduction was not observed at subtelomeres lacking a *HinfI* restriction site between the telomere repeat and the region analysed by amplification (Fig. 3c and Supplementary Figs 6 and 7), indicating that the additional *HinfI* digestion *per se* did not affect DRIP efficiency. In the case of subtelomere 22q, the enzyme cocktail generates a relatively long TRF, and the additional digestion with *HinfI*, whose site is present in close proximity upstream to the amplified region, results in the release of a shorter TRF (Supplementary Fig. 7). Analysis of subtelomere 22q revealed that, following *HinfI* digestion, the hybrid enrichment levels were higher, indicating that the upstream regions, now disconnected from the shorter TRF, are not responsible for the generation of the DNA:RNA hybrids in these regions. In WT samples, we noticed an average decrease in hybrid enrichment after *HinfI* digestion in the same subtelomeres that were affected in ICF; however, in contrast to ICF samples, this decrease was not statistically significant

(Fig. 3d). The subtelomeres lacking a distal *HinfI* site failed to demonstrate consistent differences in hybrid enrichment between the *HinfI*-digested and non-digested samples (Fig. 3e). Altogether, these results indicate that, while the hybrids are predicted to initiate at subtelomeres in the vicinity of TSSs, they extend into telomere-hexameric repeats, which constitute a crucial component of the hybrids at chromosome ends.

**High TERRA levels are present in ICF cells during S phase.** A mechanism by which TERRA could form DNA:RNA hybrids concomitantly with telomere replication, leading to fork stalling and telomere loss events, has been suggested previously for high TERRA-expressing yeast cells, as well as for ICF cells[14,44]. Such a mechanism would require TERRA to be present at high levels during S phase. Measurement of TERRA levels at various cell-cycle stages in the telomerase-positive HeLa, HT1080 and SJSA1 cell lines[45–48] demonstrated a decline in TERRA levels in late S and G2. On the other hand, in ALT-positive cells, in which TERRA levels are much higher, no decline from S to G2 phases was found[46]. As shown previously[33,34] and here (Supplementary Fig. 4), TERRA levels are markedly elevated in cycling ICF cell populations; however, TERRA levels at specific cell-cycle stages in ICF cells had not been determined previously. To this end, avoiding drug-induced cell-cycle synchronization, we used live cell DNA staining (Vybrant DyeCycle Violet stain) and sorted

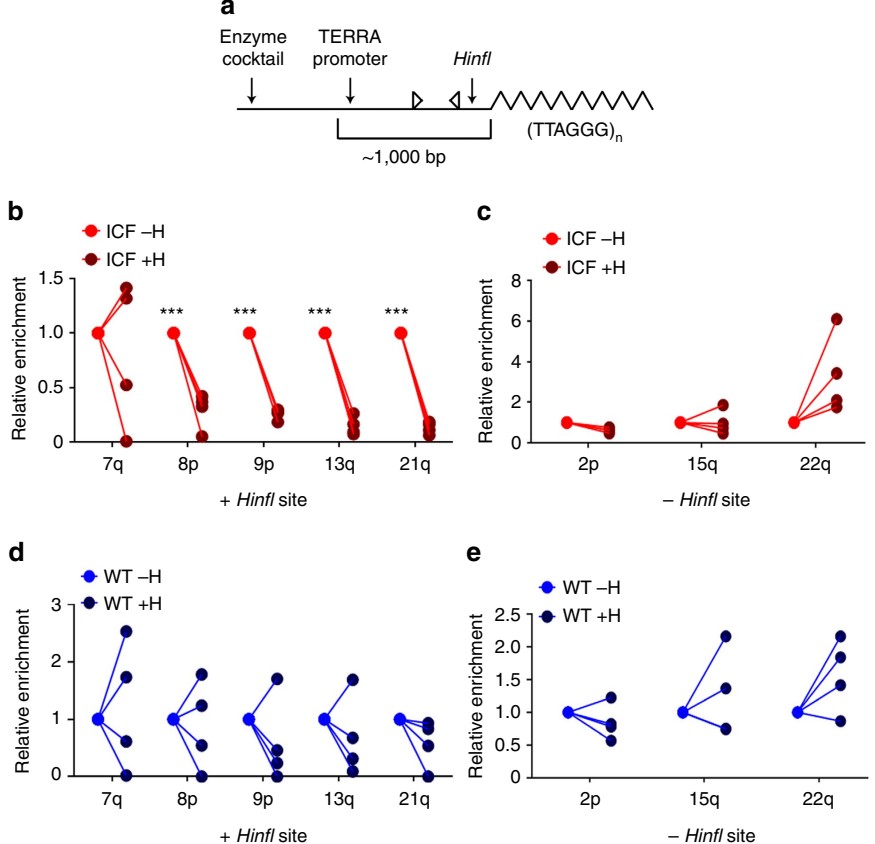

**Figure 3 | Telomere-hexameric repeats are a major component of DNA:RNA hybrids at chromosome ends.** (**a**) A schematic representation of the modified DRIP analysis. Similar to the illustration in Fig. 2a, subtelomeric sequence appears as a straight line and the telomere is depicted as a zigzagged line. The two horizontal head-to-head arrows depict the location of the PCR primers used to amplify the DRIP sample. In a subset of subtelomeres, a *HinfI* restriction site is present in close proximity to the telomere tract (right vertical arrow), downstream to the region amplified by PCR. When DNA is digested with *HinfI* in addition to the enzyme cocktail, a subtelomeric region disconnected from the telomere tract may be pulled down by DRIP. The PCR primers depicted in the scheme will amplify such fragments. (**b–e**) DNA samples of four ICF LCLs (pCor, pG, pY, pH) and four WT LCLs (GM08729, GM19116c, fY, 3125) were digested with the enzyme cocktail and then each sample was split into two fractions. One of these fractions was further digested with *HinfI*, and DRIP was performed on both fractions. The enrichment values with (+H) and without (−H) *HinfI* digestion were compared, setting the enrichment value of samples lacking the additional *HinfI* digestion as 1. Each line connects between the enrichment values of the same sample with or without additional *HinfI* digestion. (**b,d**) Analysis of subtelomeres that contain a *HinfI* restriction site in close proximity to the telomere as described in panel **a**. (**c,e**) Analyses of subtelomeres that lack a *HinfI* restriction site and serve as controls. (**b,c**) Analyses of various subtelomeres, with and without *HinfI* sites, in four ICF samples. (**d,e**) Analysis of various subtelomeres, with and without *HinfI* sites, in four WT samples. *** = $P$ values < 0.001, two-tailed Student's *t*-test.

three ICF and two WT LCLs into G1, S and G2 phases and examined fraction purity based on propidium iodide staining (Fig. 4, upper left corner). Following cell sorting, RNA was extracted from each cell-cycle fraction, and cDNA was generated using a telomere-specific oligonucleotide primer. TERRA cDNA was subjected to qRT-PCR, examining five different subtelomeres (Fig. 4). As expected, TERRA levels were higher in ICF cells for four out of the five examined subtelomeres, (with the exception of 16p), in comparison to WT ($P$ value < 0.001, Wilcoxon rank-sum test). Notably, we found that TERRA levels were consistently higher in S phase versus G1 phase for all the five studied subtelomeres in all three tested ICF cells ($P$ value < 0.001, Wilcoxon signed-rank test) while TERRA levels in the two WT cell lines demonstrated relatively consistent levels of TERRA throughout the cell cycle. The lowest difference in TERRA levels between the ICF and WT groups was detected for subtelomere 16p (Fig. 4), in agreement with the insignificant increase of hybrid formation in ICF versus WT for this subtelomere (Fig. 2d). In order to confirm that ICF LCLs indeed vary in their TERRA distribution in comparison to WT-LCLs, we analysed three additional WT-LCLs for TERRA

levels during the cell cycle. These additional WT controls similarly demonstrated relatively consistent levels of TERRA throughout the cell cycle (Supplementary Fig. 8). These findings indicate that surprisingly, in contrast to previous studies on other cell types, TERRA levels do not decline during S and G2 phases in WT LCLs. Altogether these findings demonstrate that ICF LCLs contain high levels of TERRA throughout the cell cycle, with even higher levels present in S phase supporting the scenario that telomeric DNA:RNA hybrids are present during this cell-cycle phase in ICF cells.

**Telomeric DNA:RNA hybrids form throughout the cell cycle.** We next examined whether high TERRA levels during S phase in ICF cells correspond to the presence of telomeric DNA:RNA hybrids during this cell-cycle stage. To our knowledge, cell-cycle analysis of DNA:RNA hybrids was not investigated previously. We therefore sorted three ICF and two WT LCLs, as described above, to obtain cells from G1, S and G2 phases and proceeded with DRIP on these cell-cycle fractions probing six subtelomeric regions (Fig. 5, and Supplementary Fig. 9). Enrichment values

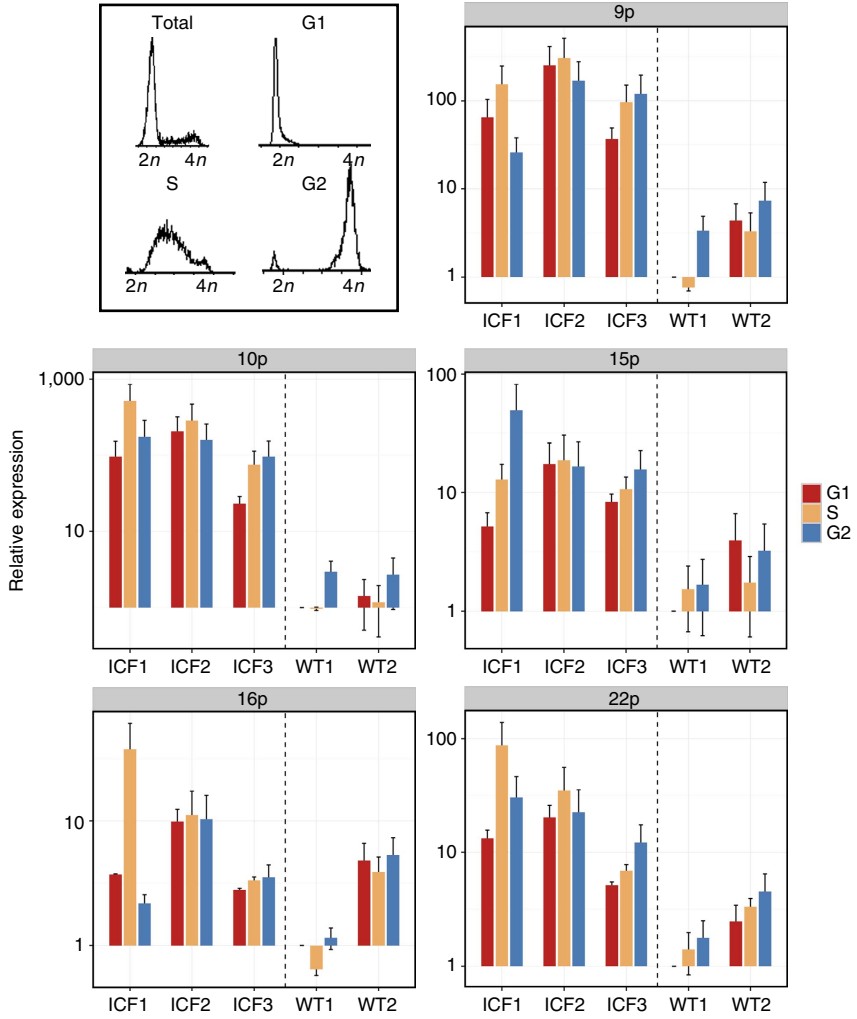

**Figure 4 | TERRA is abundant in ICF LCLs during all stages of the cell cycle.** Relative TERRA levels were determined in various LCLs sorted to G1, S and G2/M phases. FACS analysis of cell-cycle distribution prior and after sorting was performed to validate the purity of the sorted cell populations. The upper left boxed region displays a typical example of the cell-cycle distribution of an LCL population prior to sorting (depicted as 'Total') stained with vibrant dye cycle violet, and propidium iodide staining of the enriched populations of G1, S and G2/M after sorting. qRT-PCR analysis of TERRA in the sorted cell-cycle fractions was carried out for five telomeres (9p, 10q, 15q, 16p and 22q) in three ICF LCLs (pCor, pG and pY designated as ICF1, 2 and 3, respectively) and two WT LCLs (GM08729 and GM19116c designated WT1 and 2, respectively). For each subtelomere, TERRA expression in the G1 sample of WT1 was set at 1 and all other expression levels are described relatively to this sample. Bars and error bars represent means and s.e.m. of two experimental repeats. To be noted—the y axis is presented in a logarithmic scale.

obtained from G1 fractions were set at 1, and enrichment values obtained from the S and G2 fractions were compared relative to the G1 values. In agreement with the relatively high TERRA levels detected at all cell-cycle stages in ICF LCLs (Fig. 4), DNA:RNA hybrids were detected throughout the cell cycle. The enrichment in S phase was significantly higher in comparison to G1 ($P$ value $<0.05$, Wilcoxon signed-rank test) for the analysed subtelomeres in samples ICF1 and ICF2, however, not so for ICF3. In WT cells, no consistent pattern of cell-cycle stage-specific enrichment was apparent. Altogether, these finding demonstrate that telomeric DNA:RNA hybrids are present both in ICF and WT cells throughout the cell cycle with no distinct and reproducible pattern in all the examined cell lines and suggest that, when present in S-phase, these DNA:RNA hybrids could affect telomere replication.

**Telomeres in ICF cells display RNase H1-sensitive DNA damage.** Several observations suggest that DNA double-strand breaks arise from a collision between the replication machinery and an R-loop

(reviewed in refs 49 and 50). Double-strand breaks resulting from such encounters at telomeric regions could lead to telomere loss. To further explore this possibility in ICF cells, we tested whether ICF cells display DNA damage at telomeric regions. LCLs grow in suspension and are not readily amenable for 3D-interphase analysis of telomere-dysfunction-induced foci. We therefore proceeded to examine the occurrence of DNA damage signals (DDSs) at chromosome ends in mitotic cells in ICF and WT LCLs. To enrich for cells in metaphase, the LCLs were briefly treated with colcemid, then cytospun on slides and subjected to immunofluorescence with an antibody against γ-H2AX, a marker for DNA damage. DDSs were detectable both along the chromosome arms and at the extreme chromosome ends, with signals emanating from either one or from both sister chromatids (Fig. 6a,b). Scoring the percentage of γ-H2AX-positive chromosome ends (either from one or two chromatids) in three ICF LCLs and three WT LCLs indicated that ICF LCLs exhibited significantly higher DNA damage signals at these loci (Fig. 6c and Supplementary Table 2). In contrast, no significant difference in

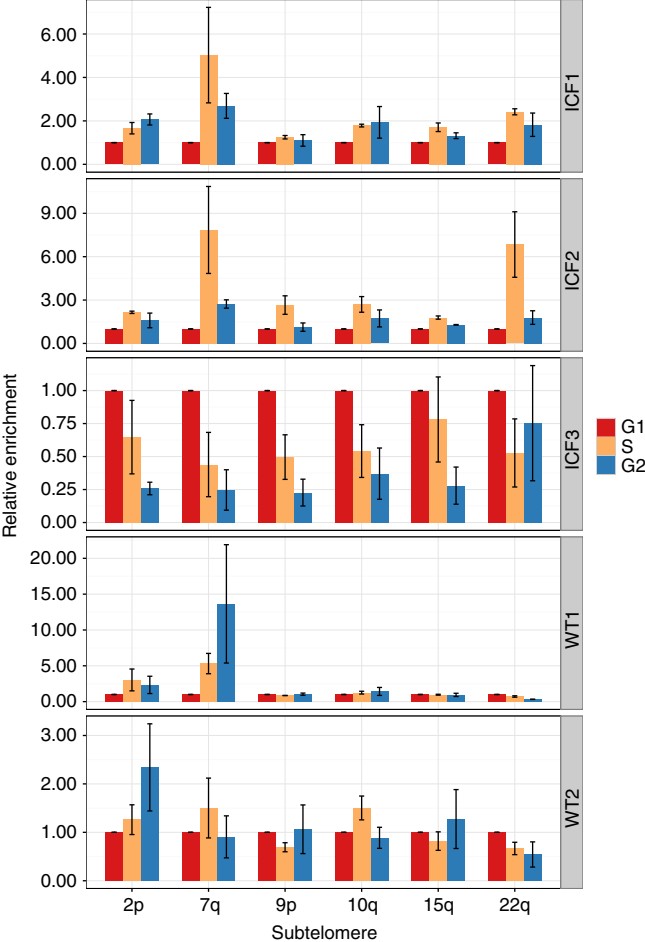

**Figure 5 | Telomeric DNA:RNA hybrids form during all stages of the cell cycle in ICF LCLs.** Three ICF LCLs (pCor, pG and pY, designated as ICF1, 2 and 3, respectively) and two WT LCLs (GM08729 and GM19116c designated WT1 and 2, respectively) were sorted to G1, S and G2 phases, after which DRIP was performed and probed for the relative enrichment of six subtelomeric regions at the various cell-cycle phases. For each subtelomere, the enrichment in G1 was set at one, and the enrichment in S and G2 was compared relatively with the G1 values. Bars and error bars represent means and s.e.m. of two experimental repeats.

the percentage of the DDS positioned along the chromosome arms was evident between the ICF and WT groups (Supplementary Fig. 10).

We next explored whether the chromosome-end DDSs are related to the observed higher levels of DNA:RNA hybrids at telomeric regions in ICF cells. To address this question, we ectopically expressed RNase H1 in two ICF and two WT LCLs. Lentiviral constructs containing RNase H1-GFP (green fluorescent protein) and a control cytoplasmic-GFP were used to infect the LCLs, and GFP-positive cells were separated by a fluorescence-activated cell sorter (FACS) and propagated in culture. We analysed the levels of RNase H1 by qRT-PCR in the sorted cells and found a eightfold increase in those expressing RNase H1-GFP in comparison to the control-GFP cells. In addition, we verified that hybrids at a non-telomeric and telomeric regions were reduced in the RNase H1-treated cells in comparison to the control-only GFP-expressing cells (Supplementary Fig. 11). Approximately a week after the infection, the cells were subjected to immunofluorescence on metaphase chromosomes to detect γ-H2AX, as described above. In WT LCLs, which display lower basal levels of chromosome-end DDS, the expression of RNase

H1 led to increased levels of chromosome-end DDS in one WT LCL, whereas the other displayed a slight reduction. However, strikingly, in both ICF LCLs, overexpression of RNase H1 resulted in a significant reduction in γ-H2AX-positive chromosome ends (Fig. 6d and Supplementary Table 3), down to the levels observed in WT cells. These findings support the notion that, in ICF cells, DNA:RNA hybrids lead to DNA damage at telomeric regions, which, if not repaired, may lead to telomere loss.

## Discussion

DNA:RNA hybrids at telomeric regions have been demonstrated previously in yeast and in human telomerase-positive and ALT cancer cells[14,28,30]. The data presented here demonstrate for the first time that DNA:RNA hybrids form at chromosome ends also in primary WT fibroblasts and in LCL lines derived from normal individuals. Using the whole-genome DRIP-seq approach, these hybrids are detected in the majority, but not in all, of chromosome ends in human fibroblasts and in the NT2 human embryonal carcinoma cell line (Fig. 1b,c). Our analysis focussed on the last 2 kb of subtelomeric regions, where TERRA promoters and TSSs have either been described[7] or are predicted to reside[40]. The presence of such hybrids in normal cells, albeit in low levels, suggests that the low TERRA levels present in normal cells are sufficient to elicit hybrid formation.

Although the human telomeric (TTAGGG)$_n$ repeat was shown by in vitro studies to form hybrids (ref. 30 and Supplementary Fig. 1), no direct in vivo evidence for the involvement of these sequences in hybrid generation was available. The protocol utilized in the previous study[30], as well as in our study, cannot discriminate whether the subtelomeric regions detected following the DRIP procedure were pulled down by the S9.6 antibody based on hybrids present in telomeric regions or in subtelomeric regions, or both. Here we demonstrate clearly that the enrichment for hybrids is significantly reduced when subtelomeric regions are subjected to DRIP following the removal of the adjacent telomeric repeats (Fig. 3). Hence, the perfectly GC-skewed telomeric repeat, which comprises the majority of the TERRA molecules, contributes greatly to the hybrids formed at chromosome ends. The stability of hybrids forming at such regions could be further enhanced by the propensity of the displaced G-rich telomeric DNA to form G-quadruplexes[51].

All mammalian chromosomes end with the canonical TTAGGG repeat, and human subtelomeres share many common characteristics, such as a high GC content, enrichment for the dinucleotide CpG, telomere-like repeats and families of larger repeats[7,8]. Despite these similarities, subtelomeric sequences vary noticeably[10]. Based on several sequence characteristics of the distal chromosome ends, we predicted that the capability to generate DNA:RNA hybrids would differ among various subtelomeres. In particular, this expectation was supported by the occurrence of GC-skewed regions, which were shown to be highly prone to hybrid formation[22], and which are present in close vicinity to the putative TERRA promoters (Fig. 1). Indeed, our findings from both whole-genome DRIP-seq data and from probing individual subtelomeres suggest that subtelomeres differ in their capacity to form hybrids (Figs 1b,c and 2). If indeed this is the case, the subtelomeric regions that comprise the 5′-end of the TERRA molecules may be the regions that 'seed' the formation of hybrids, which then could extend further into the telomere-repeat region. The difference between human telomeres in their capacity to instigate formation of DNA:RNA hybrids suggests that the integrity of specific chromosome ends may be more at risk than others. Further studies will determine whether and how a specific subtelomeric sequence can influence the function of the telomere

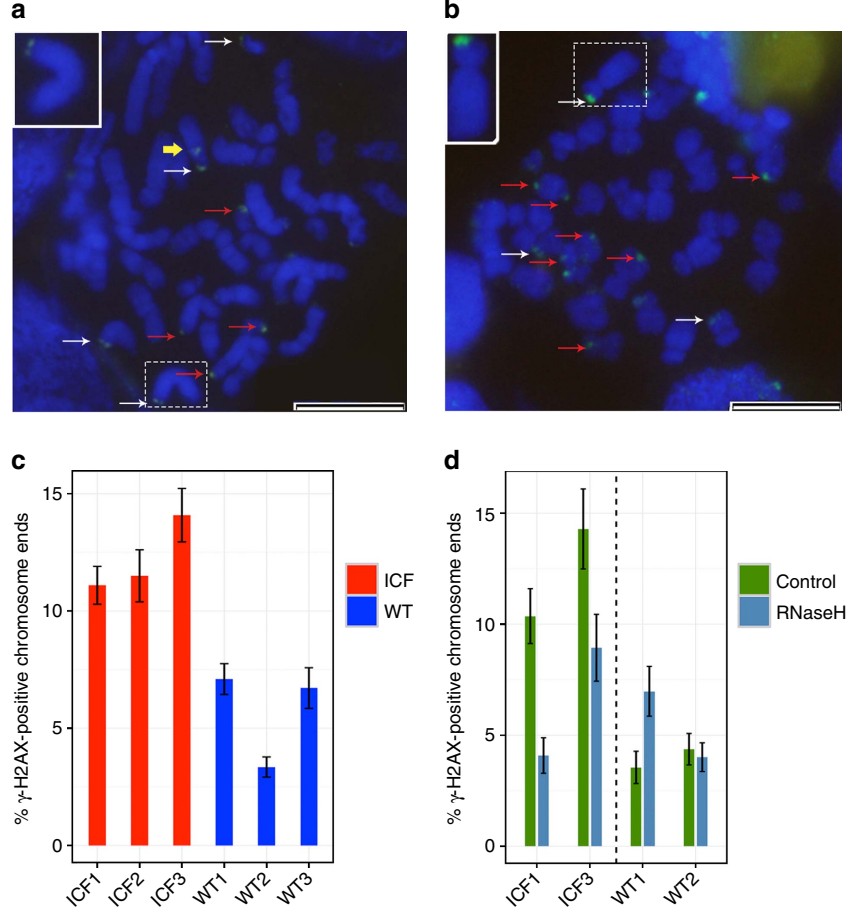

**Figure 6 | Chromosome end DNA damage signals in ICF LCLs are RNase H1 sensitive.** (**a–c**) Cytospun metaphase spreads of three ICF LCLs pCor, pG and pY (designated ICF 1, 2 and 3, respectively) and three WT LCLs GM08729, GM19116c and 3125 (designated WT 1, 2 and 3, respectively) were stained with an antibody for γ-H2AX and were then scored for the percentage of chromosome ends displaying γ-H2AX signals (either at one or both sister chromatids) (**a,b**). Representative stained metaphases of ICF1 (**a**) and ICF3 (**b**). Red arrows point to a single chromatid-stained chromosome end, white arrows point to a double-chromatid stained chromosome end and the yellow arrow points to a γ-H2AX signal positioned along the chromosome arm. The chromosome boxed by the dotted line is enlarged in the upper left corner. Bar equals 10 μM. (**c**) All three ICF samples display a significantly higher percentage of γ-H2AX signals at their chromosome ends in comparison to each of the WT samples (P value <0.001, Proportions test. All P values appear in Supplementary Table 2). At least 400 chromosomes were scored for each sample. Bars and error bars represent percentages and s.e.m. (**d**) RNase H1-GFP or cytoplasmic GFP were expressed in ICF1, ICF3, WT1 and WT2 LCLs, sorted based on GFP expression and analysed for chromosome end γ-H2AX signals. A minimum of 180 chromosomes was scored for each sample. 'RNaseH' refers to cells expressing the fused RNase H1-GFP. 'Control' refers to cells expressing the cytoplasmic-GFP. The levels of γ-H2AX signals at chromosome ends were significantly decreased in the ICF samples expressing RNase H1-GFP (P value <0.05 Proportions test. All P values appear in Supplementary Table 3). Bars and error bars represent percentages and s.e.m.

in *cis* and whether in physiological aging or in other situations where TERRA is dysregulated, such as in ICF syndrome, human telomeres differ in their propensity to shorten.

ICF type I cells are severely hypomethylated at subtelomeric regions, display a short telomere phenotype and express TERRA at extremely high levels[33]. Previous studies have demonstrated that TERRA is upregulated at short telomeres[15] and the scenario that the short telomere length in ICF syndrome influences TERRA levels cannot be ruled out. However, the high TERRA levels are an inherent characteristic of these cells independent of telomere length, as high TERRA persists in ICF cells also when telomeres are substantially elongated either by ectopic expression of hTERT or by reprogramming to iPSCs[33,37]. As TERRA is the main candidate RNA for forming DNA:RNA hybrids at subtelomere/telomeric regions, we studied whether such hybrids will form preferentially in ICF cells. However, the involvement of additional telomeric transcript species[52] in telomeric hybrids cannot be excluded. Indeed, formation of DNA:RNA hybrids at human telomeric regions is enhanced in ICF LCLs, and clearly,

the group of subtelomeres that differs most substantially in their TERRA levels between ICF and WT LCLs are those that also display significantly elevated hybrid formation in ICF cells. We cannot preclude, however, that additional characteristics of the subtelomeric and telomeric region such as the hypomethylation or aberrant chromatin modifications[34] contribute to hybrid formation. Accelerated telomere shortening in ICF syndrome could be a direct consequence of the subtelomeric/telomeric DNA:RNA hybrids, as unscheduled collisions between replication and transcription machinery have been shown to evoke double-strand breaks via DNA:RNA hybrid formation[49,53]. We postulate that, in the scenario of abnormally elevated TERRA levels, particularly during the phase at which telomeres are replicated, RNase H and additional factors that normally handle the levels of DNA:RNA hybrids would fail to deal with the hybrid overload. This would result in the accumulation of telomeric hybrids, interference with the telomere replication process and generation of DNA damage. To this end, we demonstrate that ICF LCLs display significantly higher levels of TERRA at several

chromosome ends throughout the cell cycle, especially during S phase, in comparison to WT LCLs (Fig. 4). Correspondingly, telomeric hybrids are detected in ICF cells during all the cell-cycle phases (Fig. 5). In two of the ICF LCLs that consistently showed higher levels of TERRA (ICF1 and ICF2), the enrichment of hybrids is even higher in S phase relative to G1. Excessive telomeric hybrids are expected to trigger DNA damage, and in agreement with this prediction, we demonstrate that indeed DNA damage occurs at higher levels at chromosome ends in ICF LCLs (Fig. 6a–c and Supplementary Table 2). To further support the involvement of hybrids in the chromosome end DNA damage, we ectopically expressed RNase H1 in ICF and WT cells. Strikingly, this leads to a reduction in telomeric hybrids concomitantly with significantly decreased chromosome-end DNA signals (Fig. 6d and Supplementary Table 3). Although this is most likely due to telomeric DNA:RNA hybrid degradation, we cannot exclude that RNase H1 also suppresses telomeric damage through a yet unexplored function. Altogether, our findings strongly support the hypothesis that, in ICF syndrome, TERRA is the culprit that evokes damage at telomeres by engaging in DNA:RNA hybrid formation[49,53,54]. Further studies will elucidate whether additional telomeric transcripts evoke hybrids, as well as when precisely during the cell cycle telomeric hybrids are formed. Nevertheless, ICF syndrome belongs to the emerging group of human diseases[55] in which dysregulation of a long non-coding RNA is strongly associated with an abnormal phenotype.

Cancer cells that employ the ALT pathway to maintain telomeres are also hypomethylated at subtelomeric regions and transcribe high levels of cell-cycle-dysregulated TERRA[46,56–59]. In ALT cells, the high TERRA levels are suggested to contribute to the elevated levels of hybrids, and accumulation of telomeric hybrids could lead to the homologous recombination typical to these cells[30]. ICF telomeres, contrary to ALT cancer cells, do not engage in recombination[33]. These different outcomes of high TERRA and telomeric hybrids may relate to the fact that ICF cells are untransformed and normally suppress telomere recombination, in contrast to ALT cells that carry mutations that unleash telomere recombination[46,54,60].

Regardless of the outcome, abnormally high levels of TERRA and the resulting enrichment of hybrids at telomeric regions are likely to endanger the integrity of human telomeres. Future studies that manipulate TERRA levels in various cell types, including ICF cells, will further establish the causative role of TERRA and the mechanisms by which telomeric DNA:RNA hybrids impede the normal maintenance of human telomeres.

## Methods

**Cell lines and cell culture.** ICF LCLs pCor, pG, pH, pT and pY[61] (designated as ICF1, 2, 4, 5 and 3, respectively) were grown in RPMI supplemented with 20% FCS, 2 mM glutamine, 100 U ml$^{-1}$ penicillin and 100 mg ml$^{-1}$ streptomycin. Control WT LCLs included the following: 3125 and 3133 (ref. 62), GM08729, GM08728 (ref. 33) and GM119116c (designated as WT 3, 7, 1, 6 and 2, respectively), obtained from Coriell Cell Repositories, and fY and mY LCLs generated from blood samples obtained with consent from the parents of patient pY (designated as WT 4 and 5 respectively). All control WT LCLs were grown in similar media as above, supplemented with 15% FCS. The S9.6-producing hybridoma cell line (HB-8730) was purchased from ATCC (Manassas, VA, USA) and grown in DMEM supplemented with 10% FCS, 2 mM L-Glutamine Solution and 100 mM Sodium Pyruvate.

**Plasmids.** The telomere-repeat containing plasmids were prepared as following: A 240 bp *PstI* fragment containing 40 telomere repeats was released from the pHuR93 (ATCC 61076) plasmid[63] and cloned into the pBlueScript vector, generating pBS-240bpTEL. An 810 bp *HindIII-KpnI* fragment containing 135 telomere repeats was released from pSXneo 270(T2AG3) (Addgene plasmid no. 12403, a gift from Titia de Lange[64]) and cloned into the pFC53 plasmid generating pFC53-800bpTel. A pFC53 plasmid containing mAIRN served as a positive control for DNA:RNA hybrids (pFC53-mAIRN[22]).

An RNase H1-expressing plasmid was prepared by amplification of a fragment containing the human RNase H1 starting at M27, fused to enhanced GFP (eGFP), from the pEGFP-M27-H1 plasmid (a gift from Robert Crouch[65]. The primers used for amplification were: Forward—5′-CTCAGATCTCGAGCTCAAGC-3′, containing a *BglII* restriction site (underlined) and Reverse—5′-AGTCGACTTT ACTTGTACAGCTCG-3′, containing a *SalI* restriction site (underlined). The approximately 1.58 kb amplified fragment was TA-cloned into pCR2.1 (Invitrogen). Sanger sequencing confirmed the sequence of a pCR2.1-RNase HI-eGFP-positive clone. The pTK-H1-eGFP plasmid was generated by subcloning the *BglII-SalI* fragment from the pCR2.1-RNase HI-eGFP plasmid into the *BamHI* and *XhoI* sites of the lentiviral vector pTK208, containing the cytomegalovirus promoter. Plasmid pTK113 contains the eGFP gene cloned into the pTK208 plasmid. Both pTK208 and pTK133 are gifts from Tal Kafri.

**Calculation of CpG density and GC content and GC skew.** CpG density, GC content and GC skew of 2 kb subtelomeric regions adjacent to the telomere tract were calculated using standard equations with sliding window method[23]. Heatmaps and clusters were created using heatmap.3 function from R package 'GMD'. Analysis of these regions was carried out on the GenoMatix software suite utilizing the PromoterInspector program for identification of putative Pol II promoter regions[40]. TSSs were predicted based on Nergadze *et al.*[7]

**DRIP analyses.** The S9.6 antibody was produced from the HB-8730 cell line either by recovery from ascites fluid and purification to homogeneity by Antibodies Inc. (Davis, CA, USA)[22] or as following: the HB-8730 cells were transferred to growth in serum-deprived medium (0.5% FCS in RPMI) for 48 h. Ammonium sulfate was added to the medium supernatant to 29.1 g per 100 ml, and the samples were stirred for 30 min at 4 °C. The ammonium sulfate precipitate was pelleted at 34,155g for 20 min, resuspended in a few millilitres of 10% glycerol/1 × PBS and subjected to overnight dialysis in the same buffer. The concentration and efficiency of the antibody prepared in this way were validated by comparison to the commercial S9.6 antibody (ENH001, KeraFAST).

DRIP was carried out based on ref. 22 and detailed here: Five million cells were washed in PBS, centrifuged, and resuspended without pipetting in 1.6 ml TE with 83 μl of 10% SDS and 5 μl of 20 mg per ml proteinase K and incubated overnight at 37 °C. The next morning, DNA was extracted with phenol/chloroform/ isoamylalcohol (25:24:1) (Affymetrix, no. 75831) using phase lock tubes (5prime, no. 2302840). After partial resuspension in TE, DNA was fragmented with *XhoI*, *SspI*, *HindIII*, *EcoRI* and *BsrGI* in 2.1 digestion buffer (NEB) containing 2 mM spermidine. Twenty five units of RNase H (NEB, no. M0297) were added to half of each sample to serve as a negative control, and the samples were incubated overnight at 37 °C. Following extraction in phenol/chloroform/isoamylalcohol (25:24:1) in 2 ml phase lock tubes, DNA concentration was determined using a NanoDrop *ND-1000*, and an aliquot was separated by agarose gel electrophoresis to validate efficient digestion. After setting aside 1% for input control, 4 μg of DNA were used for immunoprecipitation overnight with binding buffer (10 mM NaPO$_4$, pH 7.0/0.14 M NaCl/0.05% Triton X-100) and 10 μg of the S9.6 antibody on a rotisserie shaker at 4 °C. The following morning, 50 μl of agarose A/G beads (Pierce no. 20421) were prewashed 2 − 3 times with binding buffer and added to the DNA/antibody complex and incubated for 2 h at 4 °C on a rotisserie shaker. After three washes with binding buffer at room temperature for 10 min each, the beads were eluted with 250 μl elution buffer (50 mM Tris pH 8.0, 10 mM EDTA, 0.5% SDS) and 7 μl proteinase K (20 mg per ml) for 45 min on a rotisserie shaker at 55 °C. Finally, the eluted DNA was extracted with phenol/chloroform/ isoamylalcohol (25:24:1), precipitated with ethanol and resuspended in 50 μl of TE for analysis by qPCR.

When DRIP experiments were performed with an additional *HinfI* digestion, DNA was first digested overnight with the standard enzyme cocktail and then split into two equal aliquots. *HinfI* was added to one of the aliquots, and both aliquots were incubated at 37 °C for 3 h and then processed as described above. Primers used for qPCR reactions were designed using the Primer Express Software (Thermo Fisher Scientific) in regions as close as possible to the telomere tract, based on the reported subtelomeric sequences[10]. Owing to the repetitive nature of the subtelomeric regions, some of the primer pairs amplify more than one chromosome end. Primer sequences, their relative position to the most distal *HinfI* site and their genomic locations are described in Supplementary Table 4. All designed primers were calibrated to ensure optimal conditions for amplification. The percentage of input was calculated as $2^{(Ct1\%_{input} - Ct_{output})}$.

**DRIP-seq Mapping.** Input, untreated and bacterial RNase H-treated DRIP-seq data sets from WT human primary fibroblasts[42] and the human testicular carcinoma cell line, Ntera2 (ref. 41), were quality/adapter-trimmed using fastq-mcf[66] and mapped using Bowtie2 v2.1.0 (ref. 67) to human genome version GRCh37/hg19 supplemented by DNA sequences representing missing subtelomeric regions, when necessary. PCR duplicates were removed using SAMtools[68]. Peaks were called using MACS v2.1 (ref. 69) with input or RNase H-treated samples as controls. Fold enrichments for the most distal 2 kb of subtelomeric region of each chromosome were aggregated from peaks in that region.

**Cell sorting.** Cell sorting by DNA content was carried out as following: One microlitre of Vybrant DyeCycle Violet Stain (V35003, Life Technologies) was added to 1 ml of growth media, and the cells were stained at a concentration of 1 million cells per ml dye-containing-medium for 1–6 h at 37 °C. Cells were then sorted on a FACS Aria IIIu Cell Sorter using Violet 405 nm for excitation with a 450/40 nm bandpass filter. The irregular shape of the LCLs required the use of a 130 nm nozzle. The cells were sorted into three fractions corresponding to G1, S and G2 phases, based on the DNA content histogram. Following the sorting, 100,000 cells from each fraction were fixed in EtOH, stained with Propidium Iodide for 15 min and analysed on a FACSCalibur to validate the purity of the sorted fractions. A total of $3 \times 10^5$ and $8 \times 10^5$ cells per cell-cycle fraction were collected for RNA and DRIP experiments, respectively. Sorting of GFP-positive LCLs was carried out under aseptic conditions on the FACS Aria IIIu Cell Sorter using the 130 nm nozzle.

**RNA extraction and qRT–PCR.** ICF LCLs, whose growth is considerably attenuated, required sorting of approximately 40 million cells in order to obtain a sufficient number of cells from S and G2 phases for expression analysis. Sorted cells intended for RNA extraction were collected into 5 ml tubes containing 1 ml RNAlater solution (AM7020, Thermo Fisher Scientific). When the total volume in the tubes reached 4.5 ml, cells were centrifuged at 840g and resuspended in 1 ml RNAlater. RNA was extracted using the RNeasy Micro Kit (no. 74004, QIAGEN) and an additional DNase treatment with TURBO DNase (AM2238, Thermo Fisher Scientific) was carried out in order to eliminate any trace of DNA in the preparation. RNA concentrations were determined by Qubit fluorometric quantitation. Total RNA was reverse transcribed at 55 °C with SuperScript III reverse transcriptase (Invitrogen) using a β-actin-specific primer (5′-AGTCCG CCTAGAAGCATTTG-3′) and a TERRA-specific primer composed from five telomere-hexameric repeat ((CCCTAA)$_5$)[38]. cDNAs were then analysed with the same primers used for DRIP (Supplementary Table 4). qRT–PCR was carried out on an Applied Biosystems StepOnePlus Real-Time PCR system with Fast SYBR Green Master Mix (AB-4385612, Applied Biosystems). Analysis was carried out by the delta delta Ct method using the β-actin gene as the reference[45].

**Lentiviral infections.** Each LCL was infected serially twice with virions produced from either pTK-H1-eGFP or from pTK113 plasmids in the HEK293FT packaging cell line. Virus-containing supernatant was collected 48–72 h after transfection and polybrene was added to a final concentration of 8 μg per ml. Ten millilitres of virus-containing media were added to $5–10 \times 10^6$ cells in a 15 ml tube, and the cells were spun for 90 min at 450g at 25 °C. Following this, the cells were incubated at normal growth conditions (37 °C, 5% CO$_2$) overnight. The following day, the cells were either subjected to a second round of infection or were washed and placed in virus-free growth media for 2–3 days, after which the GFP-positive cells were collected by sorting.

**Chromosome end γH2AX staining and microscopy.** Following sorting, the GFP-positive LCLs were expanded in culture for approximately a week. γH2AX staining of metaphase spreads was based on ref. 70 and carried out as following: Prior to staining, the cells were incubated with 20 ng ml$^{-1}$ colcemid for exactly 50 min. Cells were then centrifuged for 5 min at 310g and the cell pellet was resuspended in freshly prepared hypotonic solution (0.2% KCl, 0.2% Sodium Citrate in double distilled water (DDW)) at a concentration of $5 \times 10^4$ cells ml$^{-1}$. Cells were incubated in the hypotonic solution at room temperature for 8 min exactly and cytocentrifuged for 10 min at 2000 RPM with medium acceleration on a Cytospin 3 centrifuge (SHANDON). Following fixation for 10 min in 4% paraformaldehyde in $1 \times$ PBS, slides were rinsed in water and permeabilized with KCM buffer (120 mM KCl, 20 mM NaCl, 10 mM Tris pH 7.5, 0.1% Triton) for 10 min at room temperature. The slides were blocked with 100 μg ml$^{-1}$ RNase A, 5% BSA in $1 \times$ PBS for 30 min at 37 °C. Slides were then incubated for 1 h at 37 °C with a mouse anti-phosphorylated γH2AX antibody (no. 05-636, Upstate) diluted 1:300 in 3% FCS, 0.1% triton in $1 \times$ PBS. Slides were then washed three times for 5 min in 0.1% Triton in PBS (PBST) at room temperature and incubated with a secondary antibody 1 h at 37 °C. After three washes in PBST, the slides were mounted with VECTASHIELD antifade containing 4,6-diamidino-2-phenylindole (H-1200, VECTOR LABORATORIES). Images were visualized on a BX50 microscope (Olympus), and captured with an Olympus DP73 camera controlled by the cellSens Entry Microscopy Imaging software (Olympus).

**Statistical analysis.** For each statistical analysis, we examined normality by Shapiro–Wilk Normality test and examined homoscedasticity between groups by Levene's test. When the data did not meet the assumptions of the parametric tests, we calculated the statistics by non-parametric tests as Wilcoxon signed-rank test for paired samples or the Wilcoxon signed rank-sum test (Mann–Whitney U-test) for non-paired samples.

**Data availability.** DRIP-seq data sets were published in Ginno et al.[41] and Lim et al.[42] The data sets are available in the NCBI Gene Expression Omnibus (GEO;

http://www.ncbi.nlm.nih.gov/geo/) under accession number GSE45530 and GSE57353, respectively.

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

## Acknowledgements

We thank Harold Reithman for sharing unpublished subtelomeric sequences and to Anabelle Decottignies for sharing the TERRA promoter and TSS prediction data with us and for helpful advice and comments. We thank Dale Frank and Miriam Gagliardi for helpful discussions and Daniel Kornitzer, Omer Schwartzman and Maty Tzukerman for comments on the manuscript. Thanks to the BCF unit members, Rappaport Faculty of Medicine: Yaakov Sakoury and Amir Grau from the FACS unit, and Liat Linde and Shimrat Mamrut from the genomic center for help in technical issues. This research was supported by The Israel Science Foundation (grant no. 883/12, to S. Selig), The Legacy Heritage Bio-Medical Program of the Israel Science Foundation (grant no 657/15, to S. Selig) and The National Institutes of Health (GM094299 to F.C.). S. Sagie is grateful to the Azrieli Foundation for the award of an Azrieli Fellowship. S.R.H. is a Howard Hughes Medical Institute International Student Research fellow.

## Author contributions

S. Selig, S. Sagie and F.C. conceived and designed the experiments, S. Selig, S. Sagie, S.T., H.K., A.T.-G. and S.H. performed the experiments, S. Sagie, S. Selig, F.C., S.R.H. and S.T. analysed and interpreted the data, S. Selig, S. Sagie and F.C. wrote the paper, S.T., S.R.H., C.F. and G.V. contributed to preparing the manuscript. C.F. and G.V. contributed reagents.

## Additional information

**Competing financial interests:** The authors declare no competing financial interests.

