## [Peer Review File · Nature Communications]

Reviewers' comments:

Reviewer #1 (Remarks to the Author):

The manuscript by Sagie et al. presents data on the ability of human subtelomeres/telomeres to engage in RNA:DNA hybrid formation, presumably together with the long noncoding RNA TERRA (presumably as TERRA presence in hybrids is not directly tested). In silico analysis of distal subtelomeric sequences from independent chromosome ends suggests that subtelomeres have the potential to form R-loops due to high GC content/skewing. The authors then perform S9.6 DRIPs and confirm that lymphoblasts from healthy and ICF-affected patients contain subtelomeric and telomeric R-loops, with ICF cells having higher levels of hybrids. The authors also show that TERRA is high in the S-phase of ICF cells, that telomeric R-loops exist throughout the entire cell cycle in ICF cells, and that ICF cells possibly accumulate telomere dysfunction induced foci (TIFs) containing telomeric DNA and the DDR marker gamma-H2AX. The authors conclude that aberrantly increased TERRA in ICF cells provokes elevated R-loop formation in S-phase, collision of the replication fork with R-loops, DDR activation and telomere loss.

While the subject is of clear interest, this manuscript suffers from an almost complete lack of novelty, over-interpretation of the results and a failure to properly acknowledge what has already been published. Confounding technical issues are also present.

- Contrarily to what stated several times by the authors, it is not true that in humans telomeric R-loops have been documented only in ALT cells. Arora et al. (Nature Communications 2014; cited by the authors themselves) also showed telomeric R-loops in telomerase positive cells, thus making it not surprising that the authors of this manuscript identify these structures in LCLs. The analysis shown in figure 1 is interesting, yet its novelty is only marginally incremental.

- The author's conclusion that elevated TERRA levels increase telomeric hybrids is not substantiated. Correlative evidence (higher TERRA and higher R-loops in ICF cells) is not enough. On the contrary, Arora et al. showed that induced transcription of a modified telomere in HeLa cells is not sufficient to stabilize R-loops at the same chromosome end. Thus, it seems to me that the increase in hybrids observed in ICF cells is not a direct consequence of increased TERRA but likely a consequence of deregulated CpG methylation and chromatin organization.

- The analysis of TERRA cellular levels during cell cycle progression (Figure 4) in wt cells does not fit with data previously published by the Lingner laboratory (Porro et al. Mol. Cell. Biol. 2010). I cannot see any clear drop of TERRA in S phase in wt lymphoblasts. While the authors cite the work from Porro and colleagues, they do not comment on this discrepancy. I suppose that the discrepancy stems from differences in the utilized cells. It is also possible that sorting cells into three large populations - broadly corresponding to G1, S and G2/M phases- impedes detection of changes happening in sub-phases. It is also completely unclear to me why 'the presence of high TERRA levels during S-phase supports the scenario of DNA:RNA hybrids forming during S-phase in ICF cells' (page 10); in the following paragraph the authors show that RNA hybrids are present throughout the entire cell cycle and there's no way to conclude when they form.

- The analysis of telomeric R-loops throughout cell cycle progression (Figure 5) represents the only novel aspect of this manuscript. Unfortunately, wt cells were not included in the analysis, rendering impossible to judge whether the increase of hybrids in S and G2/M unveils a defect of ICF cells or is commonly occurring in normal cells as well.

- It is not clear what the authors consider to be TIFs (Figure 6). What I see is a large cloudy signal for gamma-H2AX staining overlapped by several telomeric DNA signals. What is this? For sure not a single dysfunctional telomere.

- The conclusion that TIFs originate from hybrids is arbitrary and not substantiated. Similarly, that telomere damage and loss in ICF cells is a consequence of replication fork stalling upon collision with R-loops is farfetched. I'm surprised that no attempt was made to substantiate this hypothesis. Every laboratory working with R-loops uses over-expression of RNaseH1/2 to test to what extent the

identified dysfunctions originate from R-loops.

- That TERRA inhibits telomerase in cells is an old assumption based on in vitro data from the Lingner and Blasco laboratories. This notion has been overturned several years ago by in vivo data from independent groups (including the Lingner group). TERRA cannot be acknowledged anymore as a general telomerase inhibitor. On the contrary, recent work from the Chartrand group suggests that in budding yeast TERRA might be a POSITIVE regulator of telomerase recruitment (and thus activity) to telomeres.

Reviewer #2 (Remarks to the Author):

In this paper, Sara Selig and colleagues provide data, which support roles of TERRA in causing telomere dysfunctions in ICF (immunodeficiency, centromeric instability, facial anomalies) patients. It is demonstrated that cells from ICF patients accumulate telomeric R-loop structures. Further characterization of these R-loops indicates that they mostly involve the terminal TTAGGG-repeats whereas subtelomeric regions may contribute to a lesser extent. It is also shown that the cell cycle regulation of TERRA is lost in ICF-cells and that R-loops are retained throughout the cell cycle. R-loop persistence in S phase may perturb telomere replication and could explain telomere-loss events that occur in these patients. Consistent with this notion, DNA damage foci are increased at telomeres from ICF cells. Overall, the paper contributes to the understanding of how TERRA dysfunctions in ICF cells may impair with telomere maintenance.

Critique points:

1. For the in vitro DNA:RNA hybrid formation (Figure S1) it would be reassuring to demonstrate that sense and antisense transcription give rise to comparable amounts of RNA. In the absence of such data the increased R-loops for sense RNAs could be explained by increased transcription or by increased RNA levels rather than an increased propensity to form R-loops.
2. In the text, it seems that you refer to Figure S3 before Figure S2. Thus, the two Figures may be switched.
3. The data in Figure S3 and Figure 4 appear inconsistent. In Figure S3, the differences between wild type and ICF are much smaller (1-13 fold) than in Figure 4 (up to several 1000 fold difference for 10q TERRA).
4. Page 8: "if only hexameric repeats were responsible for hybrid formation, their levels at different chromosome ends would be expected to be similar". This statement seems only correct for equal rates of transcription, which you show does not apply to telomeres in wild type and ICF cells.
5. Figure 6: The IF/FISH data in Figure 6b are not convincing. The telomere staining seems not to stain a majority of telomeres and the gamma-H2AX foci do not co-localize with the telomere signals in a convincing manner. For this very important experiment, extensive and high quality data must be generated and shown in the figure.

Reviewer #3 (Remarks to the Author):

The paper by Sagie et al examines the formation of RNA/DNA hybrids (R-loops) at human telomeres. Using DRIP approach the authors show that R-loops are formed over telomeric and subtelomeric regions in different human cells (human primary fibroblasts and human embryonal carcinoma cells Ntera2). In addition, the authors demonstrate an enrichment of R-loops over these regions in cells

from ICF syndrome patients, a disease caused by mutation in the DNMT3B gene, involved in methylation of repetitive sequences in mammalian cells during development. The authors propose that this phenomenon is due to over-expression of TERRA RNA over these genomic regions. In addition the authors demonstrate that TERRA RNA accumulates in S cell cycle stage, correlating with the increase of R-loops, and potentially leading to telomere dysfunction and accumulation of DNA damage.

The topic of this paper is interesting, however the novelty of this paper is significantly compromised by a number of previously published high profile papers, where the existence of R-loops was demonstrated in yeast (Balk et al , NSMB 2013; Pfeiffer et al, EMBO J 2013) and human (Arora et a, Nature Comm 2014) telomeric regions. The new dimension in this paper is the analysis of subtelomeric regions (which was not done in previous publications). However, due to methodological problems the authors can not discriminate between R-loops formed in telomeric and subtelomeric regions, and therefore they fail to provide any significant data on this front (see major point 5). This paper lacks proper statistical analysis (Figures 4-6, S4 and S6 of this paper are based on single biological experiment), and therefore the meaningful interpretations can not be drawn from these figures.

Major points

1. Statistical analysis and biological replicates are missing in Figures 4-6, S2 B-C, S4 and S6. In the text the authors often refer to significant changes, when only individual experiments were shown (i.e. Fig. S2). The authors should report their data as mean +/- SEM.
2. How specific is the R-loop enrichment to telomeric regions in ICF cells? The authors should provide some non-telomeric controls where the level of R-loops are not changed in ICF cells. The R-loops data should be correlated with RNA data for the same genes.
3. Since the authors introduced the concept of GC skew, they should comment on a statistical correlation between R-loop formation as seen by DRIP-seq and GC skew in wildtype cells. Furthermore, since the authors discuss the influence of TERRA levels on hybrid formation, they should show the analysis of relative TERRA levels from different chromosome ends in wildtype cells. This discussion about TERRA levels, GC-skew and DRIP-seq signals in wildtype cells will help to understand the properties and requirements for RNA/DNA hybrid formation at subtelomeric regions.
4. In Figure 1c the DRIP-seq results obtained by the authors in previous publications (Ginno et al, MolCell 2012; Lim , eLIFE, 2015) and re-analysed in this paper, show some variability between technical replicates, e.g. NT2 exp 1(I vs II) and between biological replicates, e.g. NT2 exp1 (I) vs NT2 exp2 (I). Could the authors comment on this and offer a possible explanation? A statistical analysis of reproducibility is required to assess the quality of the DRIP-seq experiments.
5. The authors use restriction enzymes to discriminate between telomeric and subtelomeric R-loops, however in my opinion, using this methodology the authors failed to do it. First at of all the authors should provide a control for efficient digestion (i.e separation of telomeric and subtelomeric regions) which is currently missing in this paper. Secondly, the choice of HinfI seems inappropriate since it is not cutting between telomeric and subtelomeric regions of 2p, 15q, 22q with highest R-loop enrichment in ICF cells and hence, these regions can not be analysed by this method. Thirdly, the dramatic fluctuation of values after HinfI treatment for WT cells (Fig 3d) makes interpretation of results very difficult/impossible.
6. In figure 4 The WT data do not match previously published data showing a reduction of TERRA RNA in S phase (Porro et al , Nature Comm, 2014). For some probes, no difference (10q, 16p) or even an

increase in TERRA expression is detected (22q). Can the authors explain this discrepancy? In addition the presentation of Fig4 visually does not support authors' conclusions.

7. In the current version of Fig 6b, the image presented by authors does not demonstrate any clear co-localisation of gH2A.X with the telomeric signal. Indeed, the strongest signals seem to be clearly localised far away from each other. If the authors intend to claim such a correlation, they need to perform the analysis with higher resolution methods and finally by employing functional assays to affect RNA/DNA hybrids in vivo followed by the analysis of effects on DNA stability. In Figure 6c the summary of TIF results for WT cells should be included.

Respond to reviewers and to the editor

Reviewer #1

The manuscript by Sagie et al. presents data on the ability of human subtelomeres/telomeres to engage in RNA:DNA hybrid formation, presumably together with the long noncoding RNA TERRA (presumably as TERRA presence in hybrids is not directly tested). In silico analysis of distal subtelomeric sequences from independent chromosome ends suggests that subtelomeres have the potential to form R-loops due to high GC content/skewing. The authors then perform S9.6 DRIPs and confirm that lymphoblasts from healthy and ICF-affected patients contain subtelomeric and telomeric R-loops, with ICF cells having higher levels of hybrids. The authors also show that TERRA is high in the S-phase of ICF cells, that telomeric R-loops exist throughout the entire cell cycle in ICF cells, and that ICF cells possibly accumulate telomere dysfunction induced foci (TIFs) containing telomeric DNA and the DDR marker gamma-H2AX. The authors conclude that aberrantly increased TERRA in ICF cells provokes elevated R-loop formation in S-phase, collision of the replication fork with R-loops, DDR activation and telomere loss.

While the subject is of clear interest, this manuscript suffers from an almost complete lack of novelty, over-interpretation of the results and a failure to properly acknowledge what has already been published. Confounding technical issues are also present.

We thank this reviewer for his/her in-depth review of the manuscript and the many constructive remarks. Below, please find our response to the specific critiques.

1. Contrarily to what stated several times by the authors, it is not true that in humans telomeric R-loops have been documented only in ALT cells. Arora et al. (Nature Communications 2014; cited by the authors themselves) also showed telomeric R-loops in telomerase positive cells, thus making it not surprising that the authors of this manuscript identify these structures in LCLs.

We thank the reviewer for raising this point. We have corrected this and now state in the manuscript the different types of cells that have been shown to have telomeric hybrids, including telomerase-positive human cancer cells. We make it clear now in the manuscript that our novel findings are related to primary human cells and to lymphoblastoid cell lines that have not been shown previously to have hybrids.

2. The analysis shown in figure 1 is interesting, yet its novelty is only marginally incremental.

We argue that the whole-genome DRIP-seq presented in Fig. 1 contributes significantly to the field in several aspects. a. It demonstrates that telomeric hybrids are also generated in primary WT cells, and not only in cancerous cells. b. This analysis gives a global view on hybrids at almost all of the chromosome ends and therefore refutes the possibility that hybrids only form at a small subset of telomeres. This is furthermore shown to be true both for primary fibroblasts as well as for a telomerase-positive cancer cell line. c. This analysis indicates that the GC-skew at subtelomeric regions influences the levels of hybrid formation *in cis*.

3. The author's conclusion that elevated TERRA levels increase telomeric hybrids is not substantiated. Correlative evidence (higher TERRA and higher R-loops in ICF cells) is not enough. On the contrary, Arora et al. showed that induced transcription of a modified telomere in HeLa cells is not sufficient to stabilize R-loops at the same chromosome end. Thus, it seems to me that the increase in hybrids observed in ICF cells is not a direct consequence of increased TERRA but likely a consequence of deregulated CpG methylation and chromatin organization.

We now state in the manuscript that that aberrant methylation and chromatin modifications may also be factors that contribute to the propensity of telomeric hybrids in ICF syndrome. We state that while TERRA is a necessary factor for hybrid formation, it may not be the exclusive factor that influences hybrid formation.

4. The analysis of TERRA cellular levels during cell cycle progression (Figure 4) in wt cells does not fit with data previously published by the Lingner laboratory (Porro et al. Mol. Cell. Biol. 2010). I cannot see any clear drop of TERRA in S phase in wt lymphoblasts. While the authors cite the work from Porro and colleagues, they do not comment on this discrepancy. I suppose that the discrepancy stems from differences in the utilized cells. It is also possible that sorting cells into three large populations -broadly corresponding to G1, S and G2/M phases- impedes detection of changes happening in sub-phases.

Indeed, the study by Porro et al demonstrated a drop in TERRA in S-phase. However, another study did not show this reduction in TERRA during S-phase (Flynn, R. L. et al. (2015) Alternative lengthening of telomeres renders cancer cells hypersensitive to ATR inhibitors. Science 347, 273). We agree with the reviewer that this discrepancy may be the result of the different cell types analyzed or the different methods used to analyze the cell-cycle phases in relation to TERRA expression. We now remark on this discrepancy in the manuscript on page 10: “Measurement of TERRA levels during the cell cycle stages in HeLa and in various ALT cells following cell cycle synchronization (Porro et al., 2010; Flynn et al., 2015), yielded conflicting findings dependent perhaps on cell type and on cell synchronization and TERRA measurement techniques.” We conclude after analysis of 3 ICF LCLs and two WT LCLs that “we found that TERRA levels were consistently higher in S-phase versus G1-phase for all five studied subtelomeres in all the three examined ICF cells (p-value <0.001, Wilcoxon signed rank test) while the two WT cells did not show a consistent cell cycle distribution of TERRA. “

5. It is also completely unclear to me why 'the presence of high TERRA levels during S-phase supports the scenario of DNA:RNA hybrids forming during S-phase in ICF cells' (page 10); in the following paragraph the authors show that RNA hybrids are present throughout the entire cell cycle and there's no way to conclude when they form.

We thank the reviewer for this comment and agree that we cannot conclude exactly when the hybrids are formed. We state now in the manuscript that “the presence of high TERRA levels during S-phase supports the occurrence of such DNA:RNA hybrids during S-phase in ICF cells.” and do not reach conclusions regarding when the hybrids are formed. The actual presence of the hybrid is what we hypothesize may induce DNA damage, and therefore when they are generated is less relevant to our working hypothesis.

6. The analysis of telomeric R-loops throughout cell cycle progression (Figure 5) represents the only novel aspect of this manuscript.

Indeed one previous study on telomeric hybrids in human cells (Arora et al., Nature Comm 2014) demonstrated and explained many aspects of this phenomenon, however the emphasis in the previous paper was on the role and regulation of telomeric hybrids in ALT cells. Telomeric hybrids have also been studied in yeast, where they were demonstrated for the first time, and we refer to these studies in our manuscript. However, our study contains many novel aspects regarding the occurrence and role of telomeric hybrids that were not analyzed in the previous papers. We stated above in the answer to point 2 several of these novel aspects in relation to the data of the DRIP-seq experiments that appear in Figure 1. In addition, we analyze 11 subtelomeres, compared to only 3 subtelomeres in the previous study by DRIP in lymphoblastoid cells that represent an additional non-cancerous cell type. We were furthermore able to differentiate between the hybrids forming in subtelomeric versus telomeric regions, and to show that some individual subtelomeres are more prone to generate

hybrids than others, in correlation with the GC-skew in the subtelomeric region. Finally, the results we have obtained contribute to our understanding of the molecular details of telomeric defects in ICF syndrome.

7. Unfortunately, wt cells were not included in the analysis, rendering impossible to judge whether the increase of hybrids in S and G2/M unveils a defect of ICF cells or is commonly occurring in normal cells as well.

We have now performed the cell cycle analysis on two WT LCLs and these data now appears in amended Figure 5. The findings of this analysis indicate that no consistent pattern of cell-cycle stage-enrichment for telomeric hybrids was evident in the WT cells.

8. It is not clear what the authors consider to be TIFs (Figure 6). What I see is a large cloudy signal for gamma-H2AX staining overlapped by several telomeric DNA signals. What is this? For sure not a single dysfunctional telomere.

Because LCLs grow in suspension, it is a tremendous technical challenge to perform TIF analysis on interphase nuclei in these cells. To overcome this technical problem, we now present data of DNA damage at telomeres of ICF and WT LCLs obtained by chromosome-end-immunofluorescence analysis, which shows very clearly γ -H2AX signals at chromosome ends. This analysis shows that γ -H2AX signals at chromosome ends in ICF cells are elevated significantly in comparison to WT-LCLs. These results appear now in amended Figure 6.

8. The conclusion that TIFs originate from hybrids is arbitrary and not substantiated. Similarly, that telomere damage and loss in ICF cells is a consequence of replication fork stalling upon collision with R-loops is farfetched. I'm surprised that no attempt was made to substantiate this hypothesis. Every laboratory working with R-loops uses over-expression of RNaseH1/2 to test to what extent the identified dysfunctions originate from R-loops.

We greatly appreciate this critique, and have performed the experiment the reviewer suggests. We overexpressed RNaseH1 in both ICF and WT LCLs, and indeed found that in the ICF cells expressing RNaseH1 the levels of γ -H2AX signals at chromosome ends were reduced significantly. Thus the results of this experiment strongly support our working hypothesis that the telomeric hybrids influence the degree of DNA damage at telomeric regions. Regarding the hypothesis that this is due to replication-fork stalling – we do not state this as a fact but rather as a possible explanation that has also been suggested by several scientists from the telomere community (Maicher, et al., *RNA Biology*, 2012 ; Azzalin and Lingner, *Trends in Cell Biology*, 2015).

9. That TERRA inhibits telomerase in cells is an old assumption based on in vitro data from the Lingner and Blasco laboratories. This notion has been overturned several years ago by in vivo data from independent groups (including the Lingner group). TERRA cannot be acknowledged anymore as a general telomerase inhibitor. On the contrary, recent work from the Chartrand group suggests that in budding yeast TERRA might be a POSITIVE regulator of telomerase recruitment (and thus activity) to telomeres.

We removed this from the discussion

Reviewer #2:

In this paper, Sara Selig and colleagues provide data, which support roles of TERRA in causing telomere dysfunctions in ICF (immunodeficiency, centromeric instability, facial anomalies) patients. It is demonstrated that cells from ICF patients accumulate telomeric R-loop structures. Further characterization of these R-loops indicates that they mostly involve the terminal TTAGGG-repeats whereas subtelomeric regions may contribute to a lesser extent. It is also shown that the cell cycle regulation of TERRA is lost in ICF-cells and that R-loops are retained throughout the cell cycle. R-loop persistence in S phase may perturb telomere replication and could explain telomere-loss events that occur in these patients. Consistent with this notion, DNA damage foci are increased at telomeres from ICF cells. Overall, the paper contributes to the understanding of how TERRA dysfunctions in ICF cells may impair with telomere maintenance.

We thank the reviewer for his/her thorough review and his constructive remarks. We respond below to the various points brought up.

Critique points:

1. For the in vitro DNA:RNA hybrid formation (Figure S1) it would be reassuring to demonstrate that sense and antisense transcription give rise to comparable amounts of RNA. In the absence of such data the increased R-loops for sense RNAs could be explained by increased transcription or by increased RNA levels rather than an increased propensity to form R-loops.

We thank the reviewer for suggesting this important control. We now show in Figure S1 that the transcription from the sense direction is either equal to or lower than the transcription from the antisense direction. Interestingly, Arora et al., (*Nat Comm*, 2014) show a similar finding when transcribing TERRA *in vitro* from plasmids containing a 800bp long telomere tract. As we find with a similar plasmid - the yields of TERRA-like RNA were much lower than antisense-TERRA.

2. In the text, it seems that you refer to Figure S3 before Figure S2. Thus, the two Figures may be switched.

Thank you for pointing this out to us. We have double checked this time that the figures are in the correct order.

3. The data in Figure S3 and Figure 4 appear inconsistent. In Figure S3, the differences between wild type and ICF are much smaller (1-13 fold) than in Figure 4 (up to several 1000 fold difference for 10q TERRA).

We have addressed this critique in the following manner:

- a. We have repeated the DRIP experiments that appear in Figures 4 and S3 (in the revised manuscript – Fig. S4) and new graphs now display means and SDs. The differences observed between both experiments are lower now. However there are still differences in the enrichments in certain telomeres between both experiments for reasons that appear in b. and c.
- b. Figure 4 presents enrichment following cell sorting. In this type of experiment, RNA is extracted from relatively few cells, leading to fluctuations in the enrichments. Importantly, in all TERRA cycle experiments the trend was consistent, with enrichments from ICF cells being higher than those of WT cells.
- c. An additional reason for the inconsistencies seen between the data in the two previous figures has to do with the fact that in Fig. S4 we combined the TERRA levels from all the controls together and all the ICF cells together and showed the mean TERRA levels. The

TERRA levels vary between individuals (perhaps linked to the individual DNMT3B mutation), even though as a group there is a significant difference between ICF and WT cells. We now show the distribution of the results in S4 as a boxplot.

4. Page 8: "if only hexameric repeats were responsible for hybrid formation, their levels at different chromosome ends would be expected to be similar". This statement seems only correct for equal rates of transcription, which you show does not apply to telomeres in wild type and ICF cells.

We agree with this critique and have removed this sentence from the revised manuscript.

5. Figure 6: The IF/FISH data in Figure 6b are not convincing. The telomere staining seems not to stain a majority of telomeres and the gamma-H2AX foci do not co-localize with the telomere signals in a convincing manner. For this very important experiment, extensive and high quality data must be generated and shown in the figure.

Because LCLs grow in suspension, it is a tremendous technical challenge to perform TIF analysis on interphase nuclei in these cells. To overcome this technical problem, we now present data of DNA damage at telomeres of ICF and WT LCLs obtained by chromosome-end-immunofluorescence analysis, which shows very clearly γ -H2AX signals at chromosome ends. This analysis shows that γ -H2AX signals at chromosome ends in ICF cells are elevated significantly in comparison to WT-LCLs. These results appear now in amended Figure 6.

Reviewer #3:

The paper by Sagie et al examines the formation of RNA/DNA hybrids (R-loops) at human telomeres. Using DRIP approach the authors show that R-loops are formed over telomeric and subtelomeric regions in different human cells (human primary fibroblasts and human embryonal carcinoma cells Ntera2). In addition, the authors demonstrate an enrichment of R-loops over these regions in cells from ICF syndrome patients, a disease caused by mutation in the DNMT3B gene, involved in methylation of repetitive sequences in mammalian cells during development. The authors propose that this phenomenon is due to over-expression of TERRA RNA over these genomic regions. In addition the authors demonstrate that TERRA RNA accumulates in S cell cycle stage, correlating with the increase of R-loops, and potentially leading to telomere dysfunction and accumulation of DNA damage.

The topic of this paper is interesting, however the novelty of this paper is significantly compromised by a number of previously published high profile papers, where the existence of R-loops was demonstrated in yeast (Balk et al, NSMB 2013; Pfeiffer et al, EMBO J 2013) and human (Arora et al, Nature Comm 2014) telomeric regions. The new dimension in this paper is the analysis of subtelomeric regions (which was not done in previous publications). However, due to methodological problems the authors can not discriminate between R-loops formed in telomeric and subtelomeric regions, and therefore they fail to provide any significant data on this front (see major point 5) addressed below. This paper lacks proper statistical analysis (Figures 4-6, S4 and S6 of this paper are based on single biological experiment), and therefore the meaningful interpretations can not be drawn from these figures. Addressed below

We thank to reviewer for a very thorough review that relates to many important points. Our point by point response appears below.

Regarding the novelty of the findings in this paper, please see our response to points 2. and 6. of reviewer 1.

Major points

1. Statistical analysis and biological replicates are missing in Figures 4-6, S2 B-C, S4 and S6. In the text the authors often refer to significant changes, when only individual experiments were shown (i.e. Fig. S2). The authors should report their data as mean +/- SEM.

All experiments mentioned in the reviewer's critique have been repeated and the statistical analysis is now represented as mean +/- SEM.

2. How specific is the R-loop enrichment to telomeric regions in ICF cells? The authors should provide some non-telomeric controls where the level of R-loops are not changed in ICF cells. The R-loops data should be correlated with RNA data for the same genes.

These data have been added to the paper and appears in Supplementary Figure S5a & b

3. Since the authors introduced the concept of GC skew, they should comment on a statistical correlation between R-loop formation as seen by DRIP-seq and GC skew in wildtype cells.

We thank the reviewer for raising this important point and we now include Supplementary Figure S3 to address this correlation. This new data unequivocally establishes that GC skew is a key determinant of R-loop signal in wild-type cells. This analysis is now mentioned in the Results section.

Furthermore, since the authors discuss the influence of TERRA levels on hybrid formation, they should show the analysis of relative TERRA levels from different chromosome ends in

wildtype cells. This discussion about TERRA levels, GC-skew and DRIP-seq signals in wildtype cells will help to understand the properties and requirements for RNA/DNA hybrid formation at subtelomeric regions.

We agree with the reviewer that this would be very useful data. We attempted to extract TERRA levels for individual subtelomeres from published poly(A)-enriched or total fibroblast RNA-seq datasets. Unfortunately, TERRA levels are so low in WT cells that it is very difficult to use these measurements to accurately correlate TERRA levels with DRIP levels in WT cells.

4. In Figure 1c the DRIP-seq results obtained by the authors in previous publications (Ginno et al, MolCell 2012; Lim, eLIFE, 2015) and re-analysed in this paper, show some variability between technical replicates, e.g. NT2 exp 1(I vs II) and between biological replicates, e.g. NT2 exp1 (I) vs NT2 exp2 (I). Could the authors comment on this and offer a possible explanation? A statistical analysis of reproducibility is required to assess the quality of the DRIP-seq experiments.

The reviewer raises an important point, which we are happy to clarify. In Figure 1c, the slight differences in peaks between replicates stem from the way DRIP-seq is conducted. In brief, after immunoprecipitating restriction fragments that carry R-loops, the fragments are sonicated to reduce their size, and sequenced. Because sonication is random, the patterns of peaks can appear slightly different from experiment to experiment. However, we do not use this information when analyzing DRIP-seq data. Instead, we simply use the read density (and its sensitivity to RNaseH treatment) as evidence that a given restriction fragment carried R-loops and call that fragment as R-loop positive (vice-versa for R-loop negative). Thus these small fluctuations in peaks are in fact meaningless. In Figure 1c, we highlighted the smallest restriction fragment considered R-loop positive located downstream of the TERRA TSS.

Aside from this technical comment, we agree that NT2 exp 1 (I), which was the first ever dataset published and was generated using much older sequencing technology, suffered from low coverage and thus appeared a little different. Because of this we removed it from Figure 1c. All other datasets are otherwise highly correlated with each other as can be seen in systematic XY density plots of the various datasets as shown now in Supplementary Figure S2. Thus we are confident that the calls made here are accurate and robust over multiple independent DRIP-seq replicates.

5. The authors use restriction enzymes to discriminate between telomeric and subtelomeric R-loops, however in my opinion, using this methodology the authors failed to do it. First of all the authors should provide a control for efficient digestion (i.e separation of telomeric and subtelomeric regions) which is currently missing in this paper.

We thank the reviewer for this comment and in order to validate that indeed *HinfI* digestion was carried out to completion, before carrying out the DRIP, we analyzed each sample prior and post *HinfI* digestion by quantitative-PCR using primers that amplify subtelomere 16p. The amplification product of this subtelomere contains a *HinfI* site. In all the examined samples following the *HinfI* digestion, the PCR amplification decreased by 70 to several hundred fold, as demonstrated now in the revised manuscript in Supplementary Figure S6.

Secondly, the choice of HinfI seems inappropriate since it is not cutting between telomeric and subtelomeric regions of 2p, 15q, 22q with highest R-loop enrichment in ICF cells and hence, these regions can not be analysed by this method.

In this experiment we analyzed all the subtelomeres that answered the following criteria: a. has a clear promoter and transcription start site, b. Contains a *HinfI* site between the telomere

tract and the unique region in the subtelomere, c. The region between the telomere tract and the TSS is long enough and subtelomere-specific enough to allow the planning of specific PCR primers. There are very few subtelomeres that answer these criteria, and these are 7q, 8p, 9p, and 21q. We analyzed also subtelomere 13q even though the region between the telomere tract and the TSS is not specific only to this subtelomere (all the details regarding the primers appear in Table S3). We analyzed 7q, 8p, 9p, 13q and 21q and four out of these five telomeres showed a reduction in relative enrichment of hybrids following digestion with *HinfI*. We agree with the reviewer that unfortunately 2p, 15q and 22q cannot be analyzed this way, but this is a limitation of DNA sequence that we cannot overcome. However these three subtelomeres, 2p, 15q and 22q, because of the lack of *HinfI* sites between the telomere tract and the subtelomere-unique region, serve as good controls for the fact that the digestion with *HinfI* did not affect the DRIP. We conclude this because no consistent differences in DRIP enrichment were evident prior and post *HinfI* digestion. Hence we believe that the group of subtelomeres that we analyzed are representative of the two different groups of telomeres; telomeres in which the subtelomere and the telomere can be separated by the *HinfI* site, and those in which this does not occur, and therefore this second group of telomeres serves as controls for the lack of effect of *HinfI* digestion per se.

Thirdly, the dramatic fluctuation of values after HinfI treatment for WT cells (Fig 3d) makes interpretation of results very difficult/impossible.

We are happy to try to clarify this point: While in Fig. 3 the data of the individual cell lines are grouped together as ICF or WT, in Fig. S6 we provide the full input percentage data of each sample separately. When observing the ICF samples - in four out of the five subtelomeres containing a *HinfI* site, there was a clear effect following digestion (all except 7q). If we study the four WT samples with regard to these four subtelomeres, in the majority of the cases (in 12 out of the 16 cases) there was a decrease in enrichment following *HinfI* restriction, and in 4 instances a slight increase. We believe that the reason for the analysis being less consistent in the WT cells is because of the lower starting point of hybrid formation in WT cells in comparison to ICF cells. We chose to present the data as we did in Fig. 3 in order to emphasize the clear and consistent decrease we found in the ICF samples, while providing the raw data for the readers in the supplementary Figure S6.

6. In figure 4 The WT data do not match previously published data showing a reduction of TERRA RNA in S phase (Porro et al , Nature Comm, 2014). For some probes, no difference (10q, 16p) or even an increase in TERRA expression is detected (22q). Can the authors explain this discrepancy?

Indeed, the study by Porro et al demonstrated a drop in TERRA in S-phase. However, another study did not show this reduction in TERRA during S-phase (Flynn, R. L. et al. (2015) Alternative lengthening of telomeres renders cancer cells hypersensitive to ATR inhibitors. Science 347, 273). We agree with the reviewer that this discrepancy may be the result of the different cell types analyzed or the different methods used to analyze the cell-cycle phases in relation to TERRA expression. We now remark on this discrepancy in the manuscript on page 10: "Measurement of TERRA levels during the cell cycle stages in HeLa and in various ALT cells following cell cycle synchronization (Porro et al., 2010; Flynn et al., 2015), yielded conflicting findings dependent perhaps on cell type and on cell synchronization and TERRA measurement techniques." We conclude after analysis of 3 ICF LCLs and two WT LCLs that "we found that TERRA levels were consistently higher in S-phase versus G1-phase for all five studied subtelomeres in all the three examined ICF cells (p-value <0.001, Wilcoxon signed rank test) while the two WT cells did not show a consistent cell cycle distribution of TERRA."

In addition the presentation of Fig4 visually does not support authors' conclusions.

We have repeated the experiment whose data appears in Fig. 4 and find consistently that TERRA levels are higher in S-phase versus G1-phase for all five studied subtelomeres, in all the three examined ICF cells. As the reviewer suggested we changed the presentation of Fig. 4 to bar plots with SEM incorporating the new experimental repeats.

7. In the current version of Fig 6b, the image presented by authors does not demonstrate any clear co-localisation of γ H2A.X with the telomeric signal. Indeed, the strongest signals seem to be clearly localised far away from each other. If the authors intend to claim such a correlation, they need to perform the analysis with higher resolution methods

Because LCLs grow in suspension, it is a tremendous technical challenge to perform TIF analysis on interphase nuclei in these cells. To overcome this technical problem, we now present data of DNA damage at telomeres of ICF and WT LCLs obtained by chromosome-end-immunofluorescence analysis, which shows very clearly γ -H2AX signals at chromosome ends. This analysis shows that γ -H2AX signals at chromosomes ends in ICF cells are elevated significantly in comparison to WT-LCLs. These results appear now in amended Figure 6.

and finally by employing functional assays to affect RNA/DNA hybrids in vivo followed by the analysis of effects on DNA stability.

We greatly appreciate this critique, and have performed the experiment the reviewer suggests (as reviewer 1 suggested as well). We overexpressed RNaseH1 in both ICF and WT LCLs, and indeed found that in the ICF cells expressing RNaseH1 the levels of γ -H2AX signals at chromosome ends were reduced significantly. Thus the results of this experiment strongly support our working hypothesis that the telomeric hybrids influence the degree of DNA damage at telomeric regions.

In Figure 6c the summary of TIF results for WT cells should be included.

WT cells are now analyzed for the occurrence of DNA damages signals at chromosome ends and compared to the levels observed in ICF cells.

Associate editor's comments

Your manuscript entitled "Formation of DNA:RNA hybrids at human telomeres depends on subtelomeric sequence and on TERRA levels" has now been seen by 3 referees, whose comments are appended below. While they find your work of potential interest, they have raised substantive concerns that in our view preclude publication of the manuscript in Nature Communications, at least in its present form.

Should further experimental data allow you to address these criticisms, particularly but not limited to the concerns raised regarding the DRIP-seq, subtelomere analysis, quality of the immunofluorescence and the inclusion of WT controls, we would be happy to look at a revised manuscript (unless something similar has been accepted at Nature Communications or appeared elsewhere in the meantime).

We are pleased to report that we have addressed all the comments of the reviewers. These include the specific requirements stated by the associate editor, to which we respond below:

- a. *DRIP-seq* – We have addressed the concerns of reviewer #3 in our answer to points 3 and 4. We address the issue of novelty of the DRIP-seq experiments in our answer to reviewer #1, point 2.
- b. *Subtelomere analysis* – An explanation to why our experiments can tell apart the hybrids formed at subtelomeric regions versus those formed at the telomeric repeats, appears in the response to reviewer #3, point 5.
- c. *Quality of the immunofluorescence* – We have addressed this point that was brought up by all three reviewers (reviewer #1, point 8; reviewer #2, point 5; reviewer #3, point 7) by detecting DNA damage signals at telomeric regions with an alternative method. We now perform IF on metaphase chromosomes with the antibody for γ -H2AX and show very clearly the occurrence of these signals at chromosome-ends.
- d. *Inclusion of WT controls* – We have now included WT controls in all experiments, including the cell cycle DRIP experiments (respond to reviewer #1, point 7) and the detection of DNA damage signals at telomeric regions (respond to reviewer #3, point 7).

Reviewers' comments:

Reviewer #1 (Remarks to the Author):

This revised version of the manuscript by Shira Sagie and co-workers has substantially improved. While they have addressed the majority of the points raised by the reviewers only by revising the text, they also present new data that were clearly missing in the first version and that add value to the story. Nevertheless, I still feel that this work, in its current form is not suitable for publication in Nature Communications as it needs a few more (yet crucial) controls and some revision of the text/interpretations.

1) The data presented in Figure 6 are way better than what shown in the first version of the manuscript. The analysis of metaTIFs is a good choice and the additional RNaseH1 over-expression experiment seems to point to the right direction (i.e. RNA:DNA hybrids damaging telomeres in ICF cells). However, the experiment is incomplete and as it stands leaves room for alternative interpretations. It is important here to use DRIPs to show that over-expression of RNaseH1 indeed suppresses telomeric hybrids. What if RNaseH1 is suppressing hybrids at non-telomeric genomic regions (genes for example) that are somehow involved in telomere stability? While demonstration of telomeric hybrid suppression upon RNaseH1 over-expression would not fully confute the possibility that I am raising, it would at least support the authors' hypothesis (and be reassuring). On the other side, if the authors were not able to demonstrate that RNaseH1 over-expression suppresses telomeric hybrids, their hypothesis would be disproved. Second, and also very importantly, the experiment needs to be complemented with over-expression of a catalytically dead RNaseH1 variant, as to exclude alternative functions associated to RNaseH1 and that do not depend on its endoribonuclease activity.

2) Page 10: the 'conflicting findings' mentioned by the authors when they refer to TERRA cell cycle regulation are in the end not so conflicting. In the cited Flynn et al. paper, the authors showed that suppressing ATRX in HeLa cells was sufficient to stabilize TERRA levels during G2/M phase, thus providing a mechanistic explanation as to why ALT cells have elevated TERRA levels throughout the entire cell cycle. Thus, those 'discrepancies' should not be generically ascribed to different cell types or synchronization protocols, but rather to intrinsic and physiologically relevant differences between ALT and telomerase positive cells.

3) Page 13: the end of the first paragraph is speculation that should be restricted to the Discussion section. The findings indicated here by the authors support the notion that hybrids lead to telomeric DNA damage and not the notion that they lead to replication fork stalling (they are looking at metaphase chromosomes). Whether replication impairment is the cause of metaTIFs remains to be determined.

4) The authors keep on saying that TERRA is an 'absolute requirement for hybrids'. Nevertheless, they have not tested the actual presence of TERRA in the hybrids that they are analyzing and other long noncoding subtelomeric and telomeric RNAs are present in eukaryotic cells (ARRET, antiARRET and ARIA, best shown in fission yeast cells). Thus the authors cannot exclude involvement of independent RNAs other than TERRA. I'd tone down the 'absolute requirement' statement.

5) Page 15: the discussion about the RNaseH1 experiment is confusing ('While these findings support... topological stress'). I would rather say that these findings clearly support the idea that RNA:DNA hybrids are the culprits that evoke damage at telomeres in ICF cells. Nevertheless, one cannot exclude that: i) hybrids containing RNAs other than TERRA are involved; ii) hybrids form outside S-phase; iii) hybrids do not form co-transcriptionally. I do not understand why the authors are now invoking transcription-associated topological stress as they have (almost - see point 1) clearly demonstrated that gH2AX accumulation depends on RNaseH1, thus (most likely - see point 1) RNA:DNA hybrids.

6) Page 8: what kind of enzyme RNaseH is should be described as soon as the enzyme is mentioned (page 6).

7) Page 13: at the end of the first paragraph of the discussion, what does 'on a regular basis' exactly mean?

Reviewer #2 (Remarks to the Author):

In the revised paper, Sagie et al. address several of the critique points I had raised but they also introduce new issues that need to be addressed. In particular, the notion of cell cycle regulation of TERRA or its potential lack in lymphoblastoid cells from wild type donors and ICF patients remains fuzzy.

First, the statement on page 10 "Measurement of TERRA levels during all cell cycle stages in HeLa and in various ALT cells following cell cycle synchronization (44,45), yielded conflicting findings dependent perhaps on cell type and on cell synchronization and TERRA measurement techniques." seems not justified. Indeed, three independent studies obtained consistent results for HeLa and HT1080 cells as testified below:

Porro et al (MCB 2010) reported for HeLa that TERRA levels are lowest in late S phase and peak in early G1.

Flynn et al. had similar results and write in their 2011 Nature paper:

"Indeed, a recent study showed that TERRA levels significantly decrease in late S phase and increase again after S phase (Porro et al. 2010). Consistently, telomeric TERRA foci declined as cells progressed from early to late S phase (Fig. 4b and Supplementary Fig. 7c, d)."

Arnoult et al. (NSMB 2010), also obtained results that are consistent with Porro and Flynn, writing:

"Furthermore, in the TERRA levels were previously reported to vary through the cell cycle, being maximal in G1 and early S phase before decreasing to reach a minimum level in late S-G2 phase^{19,35}. Accordingly, after synchronization of HT1080+TT cells with a double-thymidine block and release into the cell cycle (Supplementary Fig. 7d), we found the expected profile for TERRA molecules, quantified by either qRT-PCR (Fig. 7a) or TERRA FISH (Fig. 7b)."

On the other hand, the data in the current paper on wild type clones are inconclusive as the analysis of the two clones give different cell cycle profiles for TERRA. These data need to be cleaned up. At this stage, it is uncertain whether or not ICF lymphoblastoid cells loose cell cycle control of TERRA expression that may or may not have been present in wild type cells. This is an important issue impinging on the molecular defects in ICF patients. Perhaps, the best way to address the issue is to complement the ICF cells with wild type DNMT3b and test if this rescues the apparent loss of cell cycle control. In addition or alternatively, enough wild type clones should be analyzed to obtain significant results. At the current stage, the data on TERRA and R-loops during the cell cycle in the two wild type cell clones are inconsistent and confusing and should not be included as figures.

Reviewer #3 (Remarks to the Author):

I appreciate the improvement and controls the authors have carried out in response to my comments. However, I still have some outstanding queries to the authors, in relation to my previous comments.

1. In relation to previous comment 1: There is still no statistics presented for Figure S6 which is important, since it shows the actual, not-normalised data for Figure 3.

2. In relation to previous comment 5: Fluctuations observed in Figure S6, demonstrating Hinf I digestion, are likely to be related to very low percentage of input for RNA/DNA hybrids detected over these regions (they are ~ 0.05%-0.15% of input). In fact in previous publication by Chedin lab (Sanz, Mol Cell 2016), positive DRIP regions were defined as having 2-15% of input enrichment, while

negative regions were defined to be at 0.01-0.1% of input. Since majority of numbers in Figure S6 are in the 'negative region' range, I wonder about the real biological relevance of these data presented in normalised Figure 3. To me it looks like all these regions represent just experimental noise. This is in line with very low and fluctuating levels of TERRA RNA, as mentioned by the authors, in response to my comment 3. Equally low DRIP values are also seen in Figure S8 (which shows not-normalised values for Figure 5) and Figure 2. It is very difficult to come up with solid biological explanation for data which appears to be at the level of experimental noise as presented in the majority of the Figures in this manuscript.

Reviewers' comments:

Reviewer #1 (Remarks to the Author):

This revised version of the manuscript by Shira Sagie and co-workers has substantially improved. While they have addressed the majority of the points raised by the reviewers only by revising the text, they also present new data that were clearly missing in the first version and that add value to the story. Nevertheless, I still feel that this work, in its current form is not suitable for publication in Nature Communications as it needs a few more (yet crucial) controls and some revision of the text/interpretations.

1) The data presented in Figure 6 are way better than what shown in the first version of the manuscript. The analysis of metaTIFs is a good choice and the additional RNaseH1 over-expression experiment seems to point to the right direction (i.e. RNA:DNA hybrids damaging telomeres in ICF cells). However, the experiment is incomplete and as it stands leaves room for alternative interpretations. It is important here to use DRIPs to show that over-expression of RNaseH1 indeed suppresses telomeric hybrids. What if RNaseH1 is suppressing hybrids at non-telomeric genomic regions (genes for example) that are somehow involved in telomere stability? While demonstration of telomeric hybrid suppression upon RNaseH1 over-expression would not fully confute the possibility that I am raising, it would at least support the authors' hypothesis (and be reassuring). On the other side, if the authors were not able to demonstrate that RNaseH1 over-expression suppresses telomeric hybrids, their hypothesis would be disproved.

We thank the reviewers for this criticism and agree that it is important to demonstrate that indeed the expression of RNaseH1 in the LCLs suppresses the levels of telomeric hybrids. We therefore carried out a DRIP experiment on ICF and control LCLs with and without overexpression of RNase H1. The results show that RNase H1 over expression is accompanied with reduction of DNA:RNA hybrid levels both at control and at telomeric regions. These results appear now in Figure S11. This strongly supports the notion that the DNA damage signals visualized at chromosomes ends that are reduced following over expression of RNaseH1, are due to the presence of hybrids at these regions.

Second, and also very importantly, the experiment needs to be complemented with over-expression of a catalytically dead RNaseH1 variant, as to exclude alternative functions associated to RNaseH1 and that do not depend on its endoribonuclease activity.

We explored the literature and failed to find reports on additional functions of RNaseH1 beyond its function as an endonuclease that specifically degrades the RNA of DNA-RNA hybrids. After demonstrating the reduction in hybrids following over expression of RNaseH1, we believe that additional experiments with a catalytically dead RNaseH1 will not fundamentally contribute to the message of this manuscript.

2) Page 10: the 'conflicting findings' mentioned by the authors when they refer to TERRA cell cycle regulation are in the end not so conflicting. In the cited Flynn et al. paper, the authors showed that suppressing ATRX in HeLa cells was sufficient to stabilize TERRA

levels during G2/M phase, thus providing a mechanistic explanation as to why ALT cells have elevated TERRA levels throughout the entire cell cycle. Thus, those 'discrepancies' should not be generically ascribed to different cell types or synchronization protocols, but rather to intrinsic and physiologically relevant differences between ALT and telomerase positive cells.

We apologize for quoting and interpreting previous studies in an inaccurate manner. We now have reworded our statement regarding the previous studies on TERRA expression during the cell cycle and state that “Measurement of TERRA levels at various cell cycle stages in the telomerase positive HeLa, HT1080 and SJS1 cell lines, demonstrated a decline in TERRA levels in late S and G2. On the other hand, in ALT-positive cells, in which TERRA levels are much higher, no decline from S to G2 phases was found.”

3) Page 13: the end of the first paragraph is speculation that should be restricted to the Discussion section. The findings indicated here by the authors support the notion that hybrids lead to telomeric DNA damage and not the notion that they lead to replication fork stalling (they are looking at metaphase chromosomes). Whether replication impairment is the cause of metaTIFs remains to be determined.

We agree that that this interpretation should appear only in the discussion and have omitted it from the end of the results section, as requested by the reviewer.

4) The authors keep on saying that TERRA is an 'absolute requirement for hybrids'. Nevertheless, they have not tested the actual presence of TERRA in the hybrids that they are analyzing and other long noncoding subtelomeric and telomeric RNAs are present in eukaryotic cells (ARRET, antiARRET and ARIA, best shown in fission yeast cells). Thus the authors cannot exclude involvement of independent RNAs other than TERRA. I'd tone down the 'absolute requirement' statement.

We reworded the sentences that claim that TERRA is an absolute requirement for hybrids. Both in the “Results” and in the “Discussion” sections we removed the word “absolute”. In the “Discussion” we also mentioned that we cannot exclude that additional telomeric transcript-species are involved in hybrid formation.

5) Page 15: the discussion about the RNaseH1 experiment is confusing ('While these findings support... topological stress'). I would rather say that these findings clearly support the idea that RNA:DNA hybrids are the culprits that evoke damage at telomeres in ICF cells. Nevertheless, one cannot exclude that: i) hybrids containing RNAs other than TERRA are involved; ii) hybrids form outside S-phase; iii) hybrids do not form co-transcriptionally. I do not understand why the authors are now invoking transcription-associated topological stress as they have (almost - see point 1) clearly demonstrated that gH2AX accumulation depends on RNaseH1, thus (most likely - see point 1) RNA:DNA hybrids.

We thank the reviewer for this comment and agree that following the additional experiments we have done that demonstrate clearly that over expression of RNase H1

suppresses telomeric hybrids, the most plausible explanation for the damage at telomeres is the hybrid formation. We have changed the discussion appropriately and removed the explanation that talks about topological stress. We also added in the discussion that: “Further studies will elucidate whether additional telomeric transcripts evoke hybrids as well as when precisely during the cell cycle telomeric hybrids are formed.”

6) Page 8: what kind of enzyme RNaseH is should be described as soon as the enzyme is mentioned (page 6).

The RNase H that we used for the experiments described in Figures S1 and S5 is a bacterial RNase H. We have now specified the source of the RNase H in the manuscript on page 8 and in the legends of Figures S1 and S5. In addition we specify the source of the RNase H for the DRIP-seq experiments, on page 6 and in the “Methods” section.

7) Page 13: at the end of the first paragraph of the discussion, what does 'on a regular basis' exactly mean?

We reworded this sentence so that it is clear.

Reviewer #2 (Remarks to the Author):

In the revised paper, Sagie et al. address several of the critique points I had raised but they also introduce new issues that need to be addressed. In particular, the notion of cell cycle regulation of TERRA or its potential lack in lymphoblastoid cells from wild type donors and ICF patients remains fuzzy.

First, the statement on page 10 "Measurement of TERRA levels during all cell cycle stages in HeLa and in various ALT cells following cell cycle synchronization (44,45), yielded conflicting findings dependent perhaps on cell type and on cell synchronization and TERRA measurement techniques." seems not justified. Indeed, three independent studies obtained consistent results for HeLa and HT1080 cells as testified below: Porro et al (MCB 2010) reported for HeLa that TERRA levels are lowest in late S phase and peak in early G1.

Flynn et al. had similar results and write in their 2011 Nature paper:

"Indeed, a recent study showed that TERRA levels significantly decrease in late S phase and increase again after S phase (Porro et al. 2010). Consistently, telomeric TERRA foci declined as cells progressed from early to late S phase (Fig. 4b and Supplementary Fig. 7c, d)."

Arnoult et al. (NSMB 2010), also obtained results that are consistent with Porro and Flynn, writing: "Furthermore, in the TERRA levels were previously reported to vary through the cell cycle, being maximal in G1 and early S phase before decreasing to reach a minimum level in late S-G2 phase^{19,35}. Accordingly, after synchronization of HT1080+TT cells with a double-thymidine block and release into the cell cycle (Supplementary Fig. 7d), we found the expected profile for TERRA molecules, quantified by either qRT-PCR (Fig. 7a) or TERRA FISH (Fig. 7b)."

We apologize for quoting previous studies in an inaccurate manner. We now have reworded our statement regarding the previous studies on TERRA expression during the cell cycle and state that "Measurement of TERRA levels at various cell cycle stages in the telomerase positive HeLa, HT1080 and SJS1 cell lines, demonstrated a decline in TERRA levels in late S and G2. On the other hand, in ALT-positive cells, in which TERRA levels are much higher, no decline from S to G2 phases was found."

On the other hand, the data in the current paper on wild type clones are inconclusive as the analysis of the two clones give different cell cycle profiles for TERRA. These data need to be cleaned up. At this stage, it is uncertain whether or not ICF lymphoblastoid cells loose cell cycle control of TERRA expression that may or may not have been present in wild type cells. This is an important issue impinging on the molecular defects in ICF patients. Perhaps, the best way to address the issue is to complement the ICF cells with wild type DNMT3b and test if this rescues the apparent loss of cell cycle control. In addition or alternatively, enough wild type clones should be analyzed to obtain significant results. At the current stage, the data on TERRA and R-loops during the cell cycle in the two wild type cell clones are inconsistent and confusing and should not be included as figures.

To provide a clear picture regarding TERRA expression during the cell cycle in LCLs in general and specifically in ICF LCLs, we performed cell cycle analysis of TERRA in three additional control non-ICF cells. We now show the data for the three new controls together with the data from the first two control LCLs in Figure S8.

Now that we have analyzed 3 ICF-LCLs and 5 control LCLs we can reach clear conclusions regarding TERRA expression during the cell cycle:

In WT-LCLs, opposed to HeLa, HT1080 and ALT cell lines, there is no clear expression pattern throughout the cell cycle. In general the levels of TERRA are quite consistent throughout all stages, and clearly there is no remarkable difference in TERRA levels between G1 and S phases. Therefore it seems that inherently, TERRA has no strict cell cycle regulation in WT-LCL. Whether this difference in regulation between LCLs and the other studied cells is due to the cell type or due to the fact that these are non-cancerous cells is unclear.

On the other hand, in the three ICF-LCLs we analyzed, we see a consistent, however not always substantial, increase in TERRA during S phase. Therefore, when comparing now three ICF samples to five WT samples, the picture that emerges is that there is a change in the control of TERRA in ICF cells, such that TERRA is expressed to some degree at higher levels in S phase.

Ultimately, what we think is relevant for the present study is that in ICF LCLs, TERRA levels remain at high levels during all the cell cycle stages. Specifically in S phase – not only do we not see a decline in TERRA levels – but actually can see an elevation of TERRA to various degrees, depending on the specific cell and specific subtelomere. The presence of TERRA during S phase indicates that it may be involved at this stage in forming DNA:RNA hybrids. Indeed, data for the presence of hybrids in S-phase, as in other cell cycle stages is presented clearly in Fig. 5.

We have appropriately revised the paragraph on page 11 that describes the results of TERRA expression throughout the cell cycle. We believe that now our findings and conclusions are presented clearly and accurately.

Reviewer #3 (Remarks to the Author):

I appreciate the improvement and controls the authors have carried out in response to my comments. However, I still have some outstanding queries to the authors, in relation to my previous comments.

- 1. In relation to previous comment 1: There is still no statistics presented for Figure S6 which is important, since it shows the actual, not-normalised data for Figure 3.*

We apologize for omitting by mistake the statistical analysis for the data in Figure S6 from the previous revised manuscript. The statistical analysis for this data appears now in Figure S6 and in Table S1.

*2. In relation to previous comment 5: Fluctuations observed in Figure S6, demonstrating *Hinf*I digestion, are likely to be related to very low percentage of input for RNA/DNA hybrids detected over these regions (they are ~ 0.05%-0.15% of input). In fact in previous publication by Chedin lab (Sanz, Mol Cell 2016), positive DRIP regions were defined as having 2-15% of input enrichment, while negative regions were defined to be at 0.01-0.1% of input. Since majority of numbers in Figure S6 are in the 'negative region' range, I wonder about the real biological relevance of these data presented in normalised Figure 3. To me it looks like all these regions represent just experimental noise. This is in line with very low and fluctuating levels of TERRA RNA, as mentioned by the authors, in response to my comment 3. Equally low DRIP values are also seen in Figure S8 (which shows not-normalised values for Figure 5) and Figure 2. It is very difficult to come up with solid biological explanation for data which appears to be at the level of experimental noise as presented in the majority of the Figures in this manuscript.*

We understand the concern of the reviewer and would like to clarify a few important points:

- Regarding the fluctuations observed in previous Figure S6 - in our previous response to comment 5 we pointed out that low and fluctuating levels of hybrids are found only in WT LCLs because of the lower starting point of hybrid formation in these cells. Similarly, in our previous response to comment 3, we referred to TERRA expression in WT cells. However in ICF cells TERRA levels are very high, and in addition to this current study, this has been demonstrated previously in publications both from our group (Yehezkel et al., Hum. Mol. Genet. **17**, 2776-2789, 2008; Sagie et al., Hum. Mol. Genet. **23**, 3629-3640, 2014) and from an additional group (Deng et al., Cell Cycle **9**, 69-74, 2010). In ICF cells the enrichment results were stable, in line with the high TERRA levels.

- Regarding the levels of enrichment of hybrids – the enrichment levels are substantially influenced by the S9.6 antibody batch. The experiments carried out on the LCLs utilized antibody purified in a different manner than in the previous studies from the Chedin laboratory and therefore the enrichment levels, including for the positive controls, are lower than in the studies published previously from the Chedin laboratory. Taking this

point in account, the results in this study consistently and significantly demonstrate that hybrids are formed in ICF cells at higher levels than in WT cells. This finding repeated itself for many subtelomeres, in cells from several ICF patients in many experimental repeats. Moreover, we documented elevated hybrids in several different experiments presented in figures Figures 2, 3 and 5. In addition, we show that treatment with RNase H1 reduces the levels of hybrids at subtelomeres. Based on all the above, we feel very confident that what we are documenting is not experimental noise but the actual occurrence of hybrids at subtelomere/telomeres in human cells.

- We would also like to emphasize that hybrids in telomeres have been manifested prior to our paper in both yeast and human cells. As a matter of fact, the study of Arora et al, (2014) published in "*Nature Communications*" journal, which showed prior to us the occurrence of hybrids at human telomeric regions, demonstrated enrichment levels which were lower than in our study. However, as in our current study, the results in Arora et al., repeated themselves in several cell types.

REVIEWERS' COMMENTS:

Reviewer #1 (Remarks to the Author):

After reading the newest version of the manuscript by the Selig laboratory, I am happy to support its publication.

I would still like to mention that surfing the literature and not finding any alternative functions associated to a polypeptide besides its canonical one (in this case RNaseH1) does not prove that such alternative functions do not exist. As an act of caution, I'd suggest the authors to include a short sentence in the results section about the unlikely yet not excludable possibility that RNaseH1 could suppress telomeric damage through so far unexplored functions.

Reviewer #2 (Remarks to the Author):

The revisions of this paper are well done and therefore the work is in my opinion now suitable for publication in Nature Communications. Apart from the discovered mechanisms in ICF cells which are well-appreciated, the paper also disproves the recent claims from the Blasco-lab that TERRA would stem only from 20q (Nat Commun. 2016 Aug 17;7:12534). It may be worth pointing out this fact in order to reduce confusion in the field.

Reviewer #3 (Remarks to the Author):

In my opinion this paper, even in its revised form, does not really provide enough molecular advance to justify its publication in Nature Comm. In particular, the current title of this paper is not supported by experimental evidence as described below.

The authors demonstrate the presence of R-loops in telomeric regions (in human fibroblasts and lymphoblasts). However, this has already been shown in yeast and human cells.

The novelty of this paper lies in analysis of ICF cells, which show an increase in R-loops compared to WT cells (Fig. 2), which potentially results in increased DNA damage (Fig.6). The authors speculate that the damage is due to increased TERRA levels, however they fail to connect these two aspects experimentally. Previously the authors have already published that TERRA levels are increased in IFC cells. However, they do not show if the increased level of R-loops in IFC cells is due to increased level of TERRA (or any other RNA participating in R-loop formation), even though this claim is made twice on page 8 and pages 13 and 14-16 (discussion) and in the title of this ms. This has not been experimentally tested, which is in line with comments of Ref 1 &2.

The authors claim that the hybrids are formed over the subtelomeric regions, however, Hinf digestion analysis which separates telomeric from subtelomeric region, results in dramatic decrease of the R-loop signal (Fig.3 and S6). This experiment clearly points towards the importance of R-loops over the telomeric repeats (which is known already anyway) and relatively minor (or actually no contribution at all) from the sub-telomeric regions. There is no experimental evidence presented in this paper to support the statement that sub-telomeric regions are important for R-loop formation (apart from their natural sequence differences), as boldly claimed in the title of the paper.

New RNase H over-expression experiment presented in Fig S11 (as requested by Ref 1) show very minimal effect on the level of R-loops, therefore it is difficult to say if the reduction of DNA damage

observed in Fig 6, following RNase H over-expression, is due to R-loops or various off-target/indirect/transcriptional effects happening due to RNase H over-expression.

REVIEWERS' COMMENTS:

Reviewer #1 (Remarks to the Author):

After reading the newest version of the manuscript by the Selig laboratory, I am happy to support its publication.

I would still like to mention that surfing the literature and not finding any alternative functions associated to a polypeptide besides its canonical one (in this case RNaseH1) does not prove that such alternative functions do not exist. As an act of caution, I'd suggest the authors to include a short sentence in the results section about the unlikely yet not excludable possibility that RNaseH1 could suppress telomeric damage through so far unexplored functions.

We thank Reviewer #1 for her/his support of the revised manuscript. Following her/his comment, we added to the “Discussion” (where it seems to us more appropriate than in the “Results” section) the sentence below that appears in bold:

“To further support the involvement of hybrids in the chromosome end DNA damage, we ectopically expressed RNase H1 in ICF and WT cells. Strikingly, this leads to a reduction in telomeric hybrids, concomitantly with significantly decreased chromosome-end DNA signals (Fig. 6d and Supplementary Table 3). **While this is most likely due to telomeric DNA:RNA hybrid degradation, we cannot exclude that RNase H1 also suppresses telomeric damage through a yet unexplored function.** “

Reviewer #2 (Remarks to the Author):

The revisions of this paper are well done and therefore the work is in my opinion now suitable for publication in Nature Communications. Apart from the discovered mechanisms in ICF cells which are well-appreciated, the paper also disproves the recent claims from the Blasco-lab that TERRA would stem only from 20q (Nat Commun. 2016 Aug 17;7:12534). It may be worth pointing out this fact in order to reduce confusion in the field.

We thank Reviewer #2 for her/his support of the revised manuscript. While we agree that our work demonstrates clearly that TERRA is transcribed from several chromosome ends, we decided not to refer to the paper describing TERRA transcription exclusively from 20q. This is due to several reasons: **a.** We ourselves did not analyze TERRA transcription from this chromosome end. **b.** We determined TERRA levels in cells other than those analyzed by the Blasco laboratory. **c.** The primers that the Blasco lab used to determine TERRA at 20q are positioned at regions much more proximal than those that we used, which were downstream to the putative TERRA promoters and very close to the telomere hexameric repeat.

Reviewer #3 (Remarks to the Author):

In my opinion this paper, even in its revised form, does not really provide enough

molecular advance to justify its publication in Nature Comm. In particular, the current title of this paper is not supported by experimental evidence as described below.

The authors demonstrate the presence of R-loops in telomeric regions (in human fibroblasts and lymphoblasts). However, this has already been shown in yeast and human cells.

The novelty of this paper lies in analysis of ICF cells, which show an increase in R-loops compared to WT cells (Fig. 2), which potentially results in increased DNA damage (Fig.6). The authors speculate that the damage is due to increased TERRA levels, however they fail to connect these two aspects experimentally. Previously the authors have already published that TERRA levels are increased in IFC cells. However, they do not show if the increased level of R-loops in IFC cells is due to increased level of TERRA (or any other RNA participating in R-loop formation), even though this claim is made twice on page 8 and pages 13 and 14-16 (discussion) and in the title of this ms. This has not been experimentally tested, which is in line with comments of Ref 1 &2.

The authors claim that the hybrids are formed over the subtelomeric regions, however, Hinf digestion analysis which separates telomeric from subtelomeric region, results in dramatic decrease of the R-loop signal (Fig.3 and S6). This experiment clearly points towards the importance of R-loops over the telomeric repeats (which is known already anyway) and relatively minor (or actually no contribution at all) from the sub-telomeric regions. There is no experimental evidence presented in this paper to support the statement that sub-telomeric regions are important for R-loop formation (apart from their natural sequence differences), as boldly claimed in the title of the paper.

We have modified the title, abstract and text throughout the manuscript and put the emphasis on our findings related to ICF syndrome. We stand by our claim that chromosome ends differ in their capacity to form hybrids, but we de-emphasized the hybrids forming on the subtelomeric regions, compared to the previous version of the manuscript.

New RNase H over-expression experiment presented in Fig S11 (as requested by Ref 1) show very minimal effect on the level of R-loops, therefore it is difficult to say if the reduction of DNA damage observed in Fig 6, following RNase H over-expression, is due to R-loops or various off-target/indirect/transcriptional effects happening due to RNase H over-expression.

We added a sentence in the discussion related to this issue, as pointed out in the response to Reviewer #1. The sentence added in the “Discussion” section appears below in bold: “To further support the involvement of hybrids in the chromosome end DNA damage, we ectopically expressed RNase H1 in ICF and WT cells. Strikingly, this leads to a reduction in telomeric hybrids, concomitantly with significantly decreased chromosome-end DNA signals (Fig. 6d and Supplementary Table 3). **While this is most likely due to telomeric DNA:RNA hybrid degradation, we cannot exclude that RNase H1 also suppresses telomeric damage through a yet unexplored function.** “